# AUG-ILA: MORE TRANSFERABLE INTERMEDIATE LEVEL ATTACKS WITH AUGMENTED REFERENCES

## ABSTRACT

An intriguing property of deep neural networks is that adversarial attacks can transfer across different models. Existing methods such as the Intermediate Level Attack (ILA) further improve black-box transferability by fine-tuning a reference adversarial attack, so as to maximize the perturbation on a pre-specified layer of the source model. In this paper, we revisit ILA and evaluate the effect of applying augmentation to the images before passing them to ILA. We start by looking into the effect of common image augmentation techniques and exploring novel augmentation with the aid of adversarial perturbations. Based on the observations, we propose Aug-ILA, an improved method that enhances the transferability of an existing attack under the ILA framework. Specifically, Aug-ILA has three main characteristics: typical image augmentation such as random cropping and resizing applied to all ILA inputs, reverse adversarial update on the clean image, and interpolation between two attacks on the reference image. Our experimental results show that Aug-ILA outperforms ILA and its subsequent variants, as well as state-of-the-art transfer-based attacks, by achieving $96.99\%$ and $87.84\%$ average attack success rates with perturbation budgets 0.05 (13/255) and 0.03 (8/255), respectively, on nine undefended models.

## 1 INTRODUCTION

Recent studies (Szegedy et al., 2013; Goodfellow et al., 2015) showed that deep neural network (DNN) models are vulnerable to adversarial attacks, where perturbations are added to the clean data to fool the models in making erroneous classification. Such adversarial perturbations are usually crafted to be almost imperceptible by humans, yet causing apparent fluctuations in the model output. The effectiveness of adversarial attacks on deep learning models raises concerns in multiple fields, especially for security-sensitive applications.

Another intriguing phenomenon is that adversarial attacks can transfer across different models (Papernot et al., 2016). One explanation for this phenomenon is the overlapping decision boundaries shared by different models (Liu et al., 2017; Dong et al., 2018). In many black-box settings where the attacker has no access to the internal state of the model, such transferability can be exploited either to generate attacks from a surrogate model (Zhou et al., 2018) or to provide proper guidance to reduce the number of model queries (Guo et al., 2019).

While many methods set their goals to generate highly transferable attacks (Moosavi-Dezfooli et al., 2016; Wu et al., 2020; Guo et al., 2020), some attempt to improve the transferability of a given adversarial example by fine-tuning it. Huang et al. (2019) proposed the Intermediate Level Attack (ILA), a technique that takes an existing adversarial example as reference and boosts its black-box transferability by maximizing the projection of the intermediate feature map discrepancies between attacks. After being fine-tuned by ILA, a strong white-box attack can achieve remarkable black-box transferability, outperforming previous methods based on direct generation of transfer-based attacks (Zhou et al., 2018; Xie et al., 2019). More specifically, ILA takes three input images as references to fine-tune the attack, including a clean image, a reference attack of the clean image, and the updated image from the previous iteration. Both the clean image and reference attack remain unchanged throughout the fine-tuning process. If we view the fine-tuning process of ILA as a generalization task that attempts to make an attack effective for different models, a possible direction for improvement is to increase the diversity of the input references, such as through applying data augmentation.

| Clean image | I-FGSM + ILA | I-FGSM + Aug-ILA (Ours) |
|:---:|:---:|:---:|
| 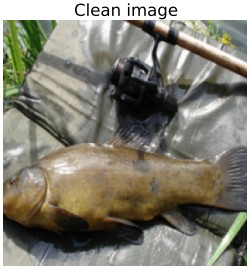 | 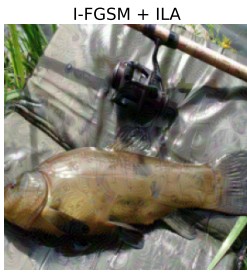 | 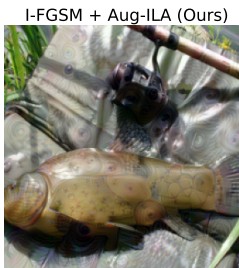 |

**Figure 1:** Visualization of the generated images among: clean image, ILA, and Aug-ILA (Ours), with the perturbation budget $\epsilon = 0.03$. Rather than adding uninterpretable noise, Aug-ILA appears to smoothen the original fine texture and overwrite new texture on the image.

In this paper, we revisit ILA and explore possible room for improvement. A natural way to promote input diversity is to apply data augmentation techniques to the input references. We first evaluate the effect of typical image transformation operations in the context of ILA. Then, we look into transformations exploiting the adversarial perturbation, and apply them together with the best image transformation found previously. Specifically, we incorporate a mixture of augmentation operations: random image cropping and resizing on all input examples, reverse adversarial update on the clean image, and attack interpolation on the reference attack. Incorporating these adaptations to the original ILA, we propose the Augmented Intermediate Level Attack (Aug-ILA) to further enhance black-box transferability. By fine-tuning simple baseline attacks such as the Iterative Fast Gradient Sign Method (I-FGSM), Aug-ILA not only outperforms ILA and its subsequent variants, but also state-of-the-art transfer-based attacks. Visualizations of the adversarial examples fine-tuned by Aug-ILA are illustrated in Figure 1 and Appendix N.

Our main contributions can be summarized as follows:

- We analyze the formulation of ILA and conduct extensive experiments to study the effect of some common image transformation operations as well as transformations exploiting the adversarial perturbation for augmenting the ILA references.

- We propose Aug-ILA, a method that fine-tunes a given adversarial attack to reinforce its attack transferability. Aug-ILA follows the framework of ILA, but with effective augmentations introduced to its references. The resulting black-box transferability of Aug-ILA outperforms the original ILA by 18.97% on average with perturbation budget $\epsilon = 0.03(\approx 8/255)$.

- We identify factors affecting the transfer success rate and explain the roles of intermediate feature perturbation and augmentation in a transfer-based attack.

## 2 RELATED WORK

In the context of adversarial machine learning, the most common threat models are the white-box and black-box settings. The white-box setting assumes that one has access to the victim model's internal state, including its gradient, parameters, training dataset, etc. The black-box setting, on the other hand, only allows querying the model with input but prohibits subsequent access to the model information. While more variations such as the grey-box and no-box settings exist, they are generally not considered in this work.

Typical white-box attacks exploit the gradient of the model to generate adversarial examples. The Fast Gradient Sign Method (FGSM) (Goodfellow et al., 2015) generates attacks by adding the signed gradient of the loss with respect to the image back to the image, obtaining a higher loss and possibly incorrect prediction. The Iterative Fast Gradient Sign Method (I-FGSM, also known as BIM) (Kurakin et al., 2017a) performs FGSM iteratively while clipping the difference between the adversarial example and the original image within the perturbation budget. Projected Gradient Descent (PGD) (Madry et al., 2018) first initiates a random start within the perturbation budget and runs I-FGSM to update the attack.

In the black-box setting, one can estimate the model information by repeatedly querying the model with varying inputs (Chen et al., 2017; Brendel et al., 2018). Recently, some novel methods have been proposed in order to reduce the number of queries (Ilyas et al., 2018; Bhagoji et al., 2018; Li et al., 2020a). However, the number of queries to the models is still abnormally large, making it easy to be detected by real-world systems. This turns the spotlight back to transfer-based attacks, where adversarial examples mainly rely on black-box transferability instead of consecutive attempts.

## 2.1 TRANSFER-BASED ATTACKS

Since strong gradient-based white-box attacks do not have high transferability by themselves (Kurakin et al., 2017b), a number of attempts have been made in the research community to improve transferability apart from increasing the attack strength.

**MI-FGSM.** Dong et al. (2018) first introduced the Momentum Iterative Fast Gradient Sign Method (MI-FGSM) to incorporate momentum to stabilize the update direction in iterative attacks, resulting in more transferable attacks.

**DIM.** The Diverse Input Method (DI$^2$-FGSM, or DIM) (Xie et al., 2019) applies random resizing and zero-padding with a certain probability to the image in each iteration of I-FGSM.

**TIM.** Dong et al. (2019) pointed out the discretion of attention map between the defended and undefended models and proposed the Translation-Invariant Method (TIM) by attacking an ensemble of translated images. To reduce the computational overhead, it was shown that kernel convolution can be applied to the gradient to achieve similar effects.

**SIM and NI-FGSM.** Lin et al. (2020) introduced the scale-invariant property of DNN and proposed the Scale-Invariant Method (SIM) to attack an ensemble of images with a scaling factor of $0.5$ on the pixel values. In the same paper, the Nesterov Iterative Fast Gradient Sign Method (NI-FGSM) was introduced, aiming to escape from poor local maxima with the Nesterov accelerated gradient.

**VMI-FGSM.** Wang & He (2021) proposed variance tuning. For each iteration, the gradient is adjusted with an expectation of gradient sampled from the image's neighborhood. Combining variance tuning with the composition of previous methods, denoted as the Composite Transformation Method (CTM), they achieved one of the strongest transfer-based black-box attacks on defended models.

A more recent work (Wu et al., 2021) replaced all input transformations with a DNN model trained to apply distortions that neutralize the adversarial noises. After preparing the denoising model, the attack is generated with the objective to sustain such distortions. All these methods suggested that adversarial transferability could be improved by augmenting the images or tuning the magnitude of gradient updates.

Alternatively, some recently proposed methods exposed the surprising nature of CNN model architectures and utilized them to generate transferable attacks. Wu et al. (2020) found that attack transferability can be enhanced by scaling up the gradient in the skip connection and referred to the technique as Skip Gradient Method (SGM). Guo et al. (2020) revised the linear property of DNN models and proposed LinBP to propagate backward without considering the non-linear layers.

## 2.2 INTERMEDIATE LEVEL ATTACK FOR FINE-TUNING ADVERSARIAL EXAMPLES

Consider an input image $\mathbf{x}$, an existing adversarial attack $\mathbf{x}'$, a model $F$, and a function $F_l$ that outputs the feature maps at layer $l$ of the model. The original ILA projection loss proposed by Huang et al. (2019) minimizes the dot product:

$$L(\mathbf{x}, \mathbf{x}', \mathbf{x}'') = -\Delta \mathbf{y}_l'' \cdot \Delta \mathbf{y}_l' \tag{1}$$

where $\Delta \mathbf{y}_l'$ and $\Delta \mathbf{y}_l''$ are two vectors defined as follows:

$$\Delta \mathbf{y}_l' = F_l(\mathbf{x}') - F_l(\mathbf{x}) \tag{2}$$

$$\Delta \mathbf{y}_l'' = F_l(\mathbf{x}'') - F_l(\mathbf{x}) \tag{3}$$

This is equivalent to maximizing $\Delta \mathbf{y}_l'' \cdot \Delta \mathbf{y}_l'$. The idea behind the formulation is to increase the norm without sacrificing the perturbation direction that causes misclassification. Due to the constraint of the perturbation budget, there is hardly room for fine-tuning in the image space. Instead, projection maximization is carried out in the feature space, specifically at layer $l$ of the model.

Based on the same framework, Li et al. (2020b) proposed another formulation of ILA:

$$\max_{\mathbf{x}''} \ (F_l(\mathbf{x}'') - F_l(\mathbf{x}))^{\mathsf{T}} \mathbf{w}^* \tag{4}$$

where $\mathbf{w}^*$ is a pre-computed parameter vector that directly maps the intermediate-level discrepancies to predict the adversarial loss, skipping the remaining model layers. One notable trick of the design is that the computation of $\mathbf{w}^*$ involves every feature map discrepancy $F_l(\mathbf{x}'_t) - F_l(\mathbf{x})$ in each iteration $t$ during the generation of $\mathbf{x}'$. We refer to this attack as ILA++ in the rest of this paper.

## 3 OUR METHOD

Based on what we have observed from the previous work, adversarial attacks tend to exhibit higher transferability when more references are involved. The increase of references can be achieved by data augmentation and reusing the temporal information over multiple iterations. From Equation (1), all three ILA input references $\mathbf{x}$, $\mathbf{x}'$ and $\mathbf{x}''$ are images, suggesting the possibility of applying image augmentation like that for representation learning to improve model generalization.

One natural way to augment the images is to apply image transformation to the input before retrieving the intermediate feature maps. Consequently, the function $F_l$ can be substituted by

$$F'_l(\mathbf{x}) = F_l(T(\mathbf{x})) \tag{5}$$

where $T$ can be any image transformation function. However, unlike attacks that enforce a wrong prediction at the logit output, ILA aims to maximize the projection of feature map discrepancies. This requires perfect pixel alignment of $\mathbf{x}$, $\mathbf{x}'$ and $\mathbf{x}''$, as a tiny shift in the image pixel induces misalignment in the feature map. Spatial transformations such as translation, cropping and rotation are required to be applied identically to all three ILA references. Otherwise, the optimization of ILA ends up being trivial in the sense of distorted feature discrepancies. Our claim is empirically verified in Appendix B, where the transformation $T$ is not identical for the images.

Different from augmentation that generalizes model representation, our task is to improve attack transferability, which can be interpreted as generalizing an adversarial example to succeed in attacking more models. Hence, apart from the common image processing methods, we may also explore any transformation associated with the adversarial perturbation. Intuitively, augmentation based on the adversarial perturbation can be viewed as an evaluation of multiple attacks before fine-tuning the final attack. This kind of pixel-wise augmentation can be applied to the images independently without conflicting with most spatial transformations. Algorithm 1 depicts the algorithmic details for applying these transformations together to ILA, where the clip function is to ensure that the first argument is within the range of the second and third arguments.

---

**Algorithm 1** Aug-ILA

**Require:** $\mathbf{x}, \mathbf{x}', F_l$, perturbation budget $\epsilon$, step size $a$, number of iterations $n$, a sequence of transformation functions $T$

    $\mathbf{x}'' \leftarrow \mathbf{x}'$
    **for** $i \leftarrow 1$ to $n$ **do**
        $\mathbf{x}_{\text{rev}} \leftarrow 2\mathbf{x} - \mathbf{x}'$                     $\triangleright$ reverse adversarial update
        Randomly initialize $T$           $\triangleright$ to ensure the image augmentations are aligned
        $\Delta\mathbf{y}'_l \leftarrow F_l(T(\mathbf{x}')) - F_l(T(\mathbf{x}_{\text{rev}}))$
        $\Delta\mathbf{y}''_l \leftarrow F_l(T(\mathbf{x}'')) - F_l(T(\mathbf{x}_{\text{rev}}))$
        $\mathbf{x}'' \leftarrow \mathbf{x}'' - a\,\mathrm{sign}(\nabla_{\mathbf{x}''}(-\Delta\mathbf{y}''_l \cdot \Delta\mathbf{y}'_l))$
        $\mathbf{x}'' \leftarrow \mathrm{clip}(\mathbf{x}'', \mathbf{x} - \epsilon, \mathbf{x} + \epsilon)$
        $\mathbf{x}'' \leftarrow \mathrm{clip}(\mathbf{x}'', 0, 1)$
        $\alpha \leftarrow \dfrac{\|\Delta\mathbf{y}''_l\|_2}{\|\Delta\mathbf{y}''_l\|_2 + \|\Delta\mathbf{y}'_l\|_2}$
        $\mathbf{x}' \leftarrow \alpha\mathbf{x}'' + (1 - \alpha)\mathbf{x}'$              $\triangleright$ attack interpolation
    **end for**
    **return** $\mathbf{x}''$

---

### 3.1 REVERSE ADVERSARIAL UPDATE ON THE CLEAN EXAMPLE

Let us reconsider Equation (2). To perform ILA, we need to obtain the feature map discrepancies between the reference attack $\mathbf{x}'$ and the unperturbed image $\mathbf{x}$. Although $\mathbf{x}$ is correctly classified

by the model, there is no information on how confident the model is and how well the input can represent the ground-truth class. In contrast, if $\mathbf{x}$ is not only correctly classified, but it is also done with a high confidence, the discrepancy of the feature maps is expected to intrinsically reflect more useful information about the attack.

In order to boost the confidence of $\mathbf{x}$, we propose to add a negative perturbation to it, which is exactly opposite to applying an adversarial attack. If we consider the entire model, this operation is expected to decrease the loss of the classifier, making the classification task easier by emphasizing features that are more significant for the task. Thus, we also expect such input to highly activate the intermediate layers, providing a better guidance for image update in ILA. Our preliminary study on the effect of reverse adversarial update can be found in Appendix A, which verifies the benign behavior brought by the reverse adversarial update. With the reverse adversarial perturbation, as expected, the model gives a lower loss and higher confidence on the correct class.

Instead of crafting a new adversarial attack which would incur extra computational demand, we can extract the perturbation from the existing reference attack $\mathbf{x}'$, turning the transformation into

$$T_{\mathrm{adv}}(\mathbf{x}) = \mathbf{x} - (\mathbf{x}' - \mathbf{x}) = 2\mathbf{x} - \mathbf{x}' \tag{6}$$

The idea of such an update is similar to DeepDream (Mordvintsev et al., 2015), in which the image is updated towards the models' interpretation of the target class. After the transformation, we look for the feature discrepancies caused by a bi-directional perturbation $\pm\epsilon\,\mathrm{sign}(\nabla_{\mathbf{x}}J(\theta, \mathbf{x}, y))$, with the positive side being the adversarial attack and the negative side being the augmentation.

## 3.2 ATTACK INTERPOLATION ON THE REFERENCE ATTACK

Besides the clean image $\mathbf{x}$, the reference attack $\mathbf{x}'$ is another factor to remain unchanged throughout all the ILA iterations. Given the image transformation in Equation (5), it is unlikely to work with another image transformation unaligned with the other images. Similar to Section 3.1, we instead exploit the adversarial noise to be the augmentation.

According to the intuition of Best Transfer Direction (BTD) in Huang et al. (2019), the stronger the reference attack is, the better ILA performs. This inspires us to strengthen the reference attack during the iterations of ILA, by interpolating the reference attack itself with a stronger attack. The output of the previous iteration, $\mathbf{x}''$, is a good candidate for a strong attack. At iteration $t$, we set

$$\mathbf{x}'_{t+1} = \alpha\mathbf{x}''_t + (1 - \alpha)\mathbf{x}'_t \tag{7}$$

where $\alpha$ is a weighting factor in the range $[0, 1]$ to control the proportion of the two attacks. This augmentation is similar to mixup (Zhang et al., 2018), except that we are mixing two adversarial examples, rather than two images from different classes. By interpolating two attacks, we can strengthen the transferability of the reference attack while preserving its original characteristics.

Based on our preliminary findings, setting $\alpha$ to non-trivial values such as $0.5$ suffices to yield satisfactory performance. However, a constant value lacks consideration of the behavior of the two attacks. Precisely, we hope to perform an adaptive interpolation depending on the effectiveness of $\mathbf{x}''_t$. If $\mathbf{x}''_t$ is weak after a single update, $\alpha$ should bias towards preserving the reference attack more than mixing with the new attack. Since ILA focuses on maximizing the linear projection between feature map discrepancies, the norm of the feature map discrepancy can be a good indicator of the performance. Consequently, we set

$$\alpha = \frac{\|\Delta\mathbf{y}''_l\|_2}{\|\Delta\mathbf{y}''_l\|_2 + \|\Delta\mathbf{y}'_l\|_2} \tag{8}$$

If we neglect the transformation in Section 3.1 for simplicity, $\Delta\mathbf{y}'_l$ and $\Delta\mathbf{y}''_l$ will remain the same as what we obtained from Equations (2) and (3), respectively. Note that the value of $\alpha$ is recomputed in each iteration before interpolation is applied.

A similar approach of reusing previous attacks in the temporal domain can be applied to the reverse adversarial update in Section 3.1. The reference attack $\mathbf{x}'$ in Equation (6) can be further extended to a collection of $\mathbf{x}'_t$ for every iteration $t$. This is not the only work to consider the temporary values of the reference attack before its convergence. ILA++ (Li et al., 2020b) also collects all losses computed during the generation of the reference attack, achieving superiority over the standard ILA. Different

from Li et al. (2020b) which uses past losses (with respect to $\mathbf{x}'$) to help update $\mathbf{x}''_t$, we make use of the past $\mathbf{x}'_t$ directly as an augmentation to enrich the information regarding feature discrepancies under different attacks.

## 4 EXPERIMENTAL RESULTS

In this section, we start by comparing the effect of different image transformations. After combining the transformations that benefit the most, we evaluate the attack transferability of Aug-ILA in comparison with previous methods based on fine-tuning as well as state-of-the-art transfer-based attack generation methods. Finally, we investigate the selection of parameters and identify influential factors to an attack's transferability.

**Setup.** Following the line of work in transfer-based attacks, we use a total of nine classification models to evaluate attack transferability, including ResNet50 (He et al., 2016), VGG19 (Simonyan & Zisserman, 2015), Inception V3 (Inc. V3) (Szegedy et al., 2016), Wide ResNet50 2x (WRN) (Zagoruyko & Komodakis, 2016), DenseNet161 (DenseNet) (Huang et al., 2017), ResNeXt101 (ResNeXt) (Xie et al., 2017), MobileNet V2 (MobileNet) (Sandler et al., 2018), PNASNet5 (PNAS-Net) (Liu et al., 2018) and SENet50 (SENet) (Hu et al., 2018). Among these models, we use ResNet50, VGG19 and Inception V3 as the source models of the attack. All of the models are pretrained on ImageNet (Russakovsky et al., 2015), with the model parameters of PNASNet[1] and SENet[2] obtained from public repositories and the remaining from Torchvision (Paszke et al., 2019). For the choice of the intermediate layer, we opt the layer 3-1 for ResNet50, layer 9 for VGG19, and layer 6a for Inception V3, where the former two have been shown to result in good performance by Li et al. (2020b). Due to the ineffectiveness of transfer-based targeted attacks (Liu et al., 2017), we mainly consider untargeted attacks with $\epsilon$ values of 0.05 ($\approx 13/255$) and 0.03 ($\approx 8/255$), under the $\ell_\infty$ norm constraint.

To measure the attack success rate, we randomly sample 5000 images from the ILSVRC2012 validation set with all images being classified correctly by the nine models. That means, the original test accuracy before the attack is 100% for every model. The default number of iterations of I-FGSM is 10 and the attack step size (learning rate) is set to $\max(\frac{1}{255}, \frac{\epsilon}{\text{no. of iterations}})$. To mount a complete attack, we first run I-FGSM for 10 iterations on the source model, and then pass the example as the reference attack to Aug-ILA for fine-tuning. We use the same model to be the source model of I-FGSM and Aug-ILA. We run Aug-ILA and other fine-tuning methods for 50 iterations in all the experiments.

### 4.1 EFFECTIVENESS OF COMMON IMAGE AUGMENTATION OPERATIONS

We select four image transformation operations that are commonly used, including translation, cropping, rotation and color jittering. To test our claim regarding the pixel misalignment causing poor transferability in ILA, we also implement an unaligned version of translation and cropping, such that the transformations with different randomized values are applied to each of the references $\mathbf{x}$, $\mathbf{x}'$ and $\mathbf{x}''$. We then report the transfer success rates on Inception V3 and VGG19 over varying perturbation budget $\epsilon$. The comparison of different transformations is shown in Appendix B.

From the result, translation and cropping yield more transferable attacks than the other transformations, especially when the perturbation budget $\epsilon$ is small. Under a critical constraint when $\epsilon = 0.01$ ($\approx 3/255$), augmentations without proper alignment fail to transfer at all, with the attack success rate very close to 0. Between translation and cropping, cropping consistently gives better performance. A further evaluation of attack transferability after the incorporation of reverse adversarial update (denoted as 'adversarial') is reported in Appendix C, demonstrating that further incorporation of adversarial reverse update leads to higher transferability. Among different combinations, we find that 'adversarial + cropping' stably gives outstanding results, while further complement of translation degrades the transferability. Hence, we opt for random cropping together with reverse adversarial update as the default augmentation of Aug-ILA.

---

[1] https://github.com/Cadene/pretrained-models.pytorch
[2] https://github.com/moskomule/senet.pytorch

text in size 8 pt

| $\epsilon$ | Method | ResNet50* | Inc. v3 | WRN | VGG19 | PNASNet |
|---|---|---|---|---|---|---|
| 0.05 | I-FGSM | **100.0%** | 25.80% | 72.12% | 48.34% | 29.62% |
| | I-FGSM + ILA | **100.0%** | 51.50% | 93.20% | 86.60% | 65.20% |
| | I-FGSM + ILA++ | 99.98% | 63.68% | 96.36% | 90.57% | 70.34% |
| | I-FGSM + Aug-ILA (Ours) | 99.78% | **90.02%** | **99.04%** | **98.76%** | **93.18%** |
| | VMI-CT-FGSM | 99.96% | 69.66% | 91.04% | 81.78% | 73.38% |
| | I-FGSM + LinBP + SGM + ILA | **100.0%** | 72.46% | 98.88% | 96.82% | 81.82% |
| 0.03 | I-FGSM | 99.96% | 14.88% | 41.28% | 26.36% | 17.40% |
| | I-FGSM + ILA | 99.98% | 34.56% | 79.88% | 66.72% | 43.32% |
| | I-FGSM + ILA++ | 99.98% | 41.44% | 87.14% | 75.24% | 49.18% |
| | I-FGSM + Aug-ILA (Ours) | 99.42% | **66.02%** | **93.92%** | **92.90%** | **75.76%** |
| | VMI-CT-FGSM | 98.28% | 47.42% | 69.94% | 56.46% | 48.08% |
| | I-FGSM + LinBP + SGM + ILA | **100.0%** | 42.08% | 90.82% | 84.08% | 53.90% |

| $\epsilon$ | Method | DenseNet | ResNeXt | MobileNet | SENet | Average |
|---|---|---|---|---|---|---|
| 0.05 | I-FGSM | 57.80% | 70.90% | 54.56% | 71.92% | 59.01% |
| | I-FGSM + ILA | 85.50% | 91.00% | 84.00% | 91.90% | 83.21% |
| | I-FGSM + ILA++ | 92.32% | 92.30% | 89.34% | 96.28% | 87.90% |
| | I-FGSM + Aug-ILA (Ours) | **98.40%** | **96.90%** | **98.12%** | 98.72% | **96.99%** |
| | VMI-CT-FGSM | 88.50% | 82.22% | 82.82% | 92.64% | 84.67% |
| | I-FGSM + LinBP + SGM + ILA | 97.64% | 95.96% | 95.60% | **98.96%** | 93.13% |
| 0.03 | I-FGSM | 31.42% | 41.82% | 31.88% | 44.00% | 38.78% |
| | I-FGSM + ILA | 69.16% | 78.48% | 67.60% | 80.16% | 68.87% |
| | I-FGSM + ILA++ | 79.08% | 78.46% | 75.28% | 87.64% | 74.83% |
| | I-FGSM + Aug-ILA (Ours) | **91.04%** | **87.76%** | **90.38%** | **93.40%** | **87.84%** |
| | VMI-CT-FGSM | 66.06% | 58.02% | 60.22% | 72.84% | 64.15% |
| | I-FGSM + LinBP + SGM + ILA | 85.24% | 80.58% | 80.56% | 92.64% | 78.88% |

* The source model used to generate the attack.

**Table 1:** Attack success rates of ImageNet adversarial examples on different models, generated from ResNet50 in the untargeted setting.

## 4.2 BLACK-BOX TRANSFERABILITY OF AUG-ILA

Our previous experiments only evaluate the transformation function $T$, without interpolation on the reference attack. In this section, we enable all augmentations discussed previously, forming the complete Aug-ILA. We first demonstrate the superiority of Aug-ILA over ILA (Huang et al., 2019) and ILA++ (Li et al., 2020b). We use ResNet50 as the source model and choose layer 3-1 as the intermediate layer for all three fine-tuned attacks. The result is shown in Table 1, while the comparison of attacks generated from VGG19 and Inception V3 can be found in Appendix E. Moreover, additional experiments with single-step attack (FGSM) and other datasets (CIFAR-10 and CIFAR-100) are shown in Appendix F and Appendix G, respectively.

In Table 1, we can see that Aug-ILA outperforms both ILA and ILA++ for all the models except the source model (ResNet50). The apparent improvement in attack transferability reflects the effectiveness of augmentation with respect to both image transformation and adversarial transformation. Another observation worth mentioning is that the attack difficulty increases according to how much the model architectures differ. For example, attacks generated from ResNet (with skip connections) tend to transfer better to models also with skip connections, such as WRN, DenseNet, ResNeXt, comparing to others such as Inception V3 and PNASNet. However, the improvement in transferability brought by Aug-ILA is more significant under such model dissimilarity, as reflected in the large gap of attack success rates between Aug-ILA and the baselines for Inception V3 and PNASNet. We believe such a phenomenon can be attributed to the extensive augmentations that enable Aug-ILA to exploit gradients with better resemblance to different architectures.

In the next step, we show that the supremacy of Aug-ILA is not limited to methods based on fine-tuning, but also other state-of-the-art transfer-based attacks. In (Wang & He, 2021), combination of the Composite Transformation Method (CTM) and variance tuning, dubbed VMI-CT-FGSM (or VMI-SI-TI-DI-FGSM more detailedly), exhibits remarkably high transferability among the state-of-the-art methods. On the other hand, Guo et al. (2020) found that LinBP works favorably together with SGM (Wu et al., 2020) and ILA, with their incorporation further advancing transferability to a new level. Using the same setting as before, we also include the attack success rates of the two combined methods into Table 1, with the hyper-parameters and other details specified in Appendix D.

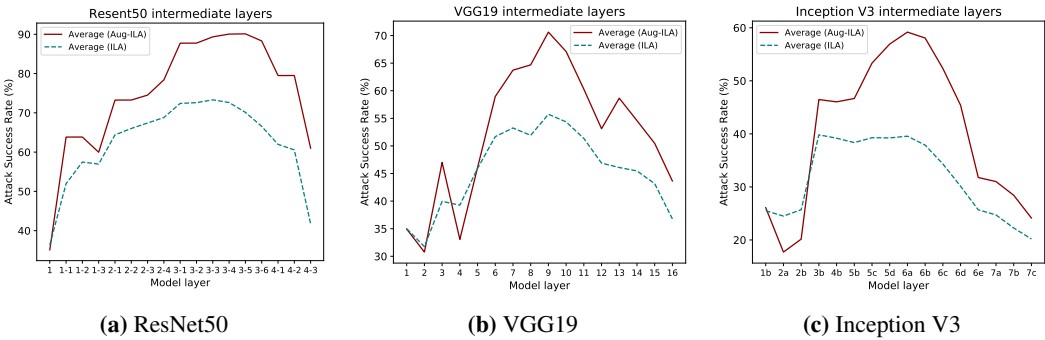

**(a)** ResNet50        **(b)** VGG19        **(c)** Inception V3

**Figure 2:** Attack transferability over different choices of the intermediate layer for ResNet50, VGG19 and Inception V3. The dotted curve shows the performance of ILA without augmentation. '3-1' in ResNet50 refers to the first residual block of the third meta-block. The naming of layers of Inception V3 follows the PyTorch convention.

Under our choices of model architecture, VMI-CT-FGSM does not work well to transfer between different model architectures, despite its remarkable performance in attacking defended models. Among the baselines, I-FGSM + LinBP + SGM + ILA achieves the best transferability between undefended models, while it is still slightly outperformed by Aug-ILA.

To further evaluate the performance of Aug-ILA, we conduct experiments on defended models. The details are reported in Appendix H.

### 4.3 EFFECT OF DIFFERENT HYPER-PARAMETERS

In the previous sections, translation and random cropping are found to be the two most effective image augmentations to be applied to the ILA references. We hope to evaluate the best values for the shifting proportion and cropping size. We test different values of the shifting factor between 0 to 0.5 inclusively for translation. Similarly, we also test with the size factor of random cropping between 0.5 and 1.0 inclusively. The experiments for both augmentations are reported in Appendix I. The result indicates that proper hyper-parameters for the transformations contribute greatly to attack transferability, with the best value of 0.05 for translation and 0.95 for random cropping.

Next, we study the effect of $\alpha$ used for attack interpolation, with fixed values between 0 and 1, in addition to the adaptive choice computed from the norm of the feature map discrepancy. The result is shown in Appendix J. Although a non-trivial value of $\alpha$ suffices to improve the performance, the adaptive selection using Equation (8) results in optimal or sub-optimal attack transferability in most cases.

Another important factor is the intermediate layer where ILA is performed. We apply ILA and Aug-ILA at each layer of the source models, and plot the attack success rates in Figure 2. The overall fluctuation pattern is similar, but Aug-ILA results in peaks that are more protruded, by virtue of the various augmentations. However, we also observe that for shallow layers, Aug-ILA turns out to have limited performance, sometimes even inferior to its baseline. While Huang et al. (2019) interpreted the tendency above in terms of the linearity of the decision boundary, we offer our explanation from the perspective of intermediate feature perturbation.

### 4.4 ROLES OF INTERMEDIATE LAYER PERTURBATION AND AUGMENTATION

Transfer-based attacks can be viewed as using one model's gradient to estimate other models' gradients. However, it was shown that the gradient directions of different models are almost orthogonal to each other (Liu et al., 2017). Locally, each layer of the target model can cause a *weakening effect* to the attack, assembling to lead to the gradient orthogonality. The deeper the attack goes, the weaker its influence becomes. For an attack on the intermediate feature, since it transits through fewer layers, both forward and backward, it is weakened to a lesser degree. One problem is, if the attack only perturbs the intermediate feature without considering the remaining layers, it is also less capable

of altering the model decision, getting demoted to an essentially weaker attack. Therefore, in the choice of the ILA layer, one that is either too shallow or too deep decays the attack transferability.

With the same insight, three factors can be identified for black-box transferability:

1. Perturbation strength
2. Similarity between the predicted gradient and the actual gradient
3. Difference in architecture between the source model and the target model.

The first two factors increase transferability while the last one hinders the attack to transfer. Factor 1 is determined by the perturbation budget, but it can be improved implicitly by ILA, such that the intermediate perturbation is also strengthened. However, factor 1 does not dominate the attack transferability, since the victim model does not necessarily have any resembling architecture to the source model. Such difference, which is factor 3, induces the *weakening effect* on the attack. We will show that Aug-ILA accomplishes a good balance between factors 1 and 2, leading to a less severe *weakening effect* from factor 3.

Since ILA maximizes the projection of feature discrepancies, the norm of the feature discrepancy should also be larger (factor 1). We visualize the L2 norm of $F_l(\mathbf{x}'') - F_l(\mathbf{x})$ at different layers in ResNet50 and compare with different attacks in Figure 3. Also, we estimate factor 2 by computing the cosine distance between the model gradient suggested by the attack $(\mathbf{x}'' - \mathbf{x})$ and the gradient direction of the clean image $(\nabla_{\mathbf{x}} J(\theta, \mathbf{x}, y))$, with the result shown in Table 2.

One interesting finding is that the L2 norm of ILA is actually stronger than that of Aug-ILA in shallow layers, which agrees with the poor transferability of Aug-ILA when applied at the first few layers in Figure 2. However, after the selected layer in ILA, the perturbation norm of Aug-ILA surpasses ILA and the situation continues until the output of the model. In the meantime, despite the strengthened feature perturbation, ILA causes the gradient direction to be more orthogonal (worse for transferability) to the actual gradient direction of the model, comparing to I-FGSM. Aug-ILA, nevertheless, slightly improves the gradient estimation without significant loss in perturbation strength. With a stronger perturbation strength together with a better predicted model gradient, Aug-ILA facilitates its comeback for the perturbation in deeper layers, and therefore its resulting attack is more transferable than the baselines.

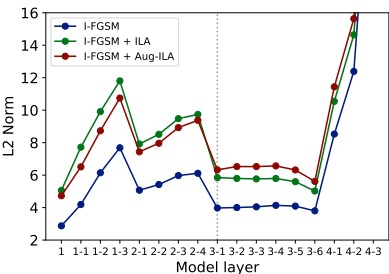

**Figure 3:** L2 norm of $F_l(\mathbf{x}'') - F_l(\mathbf{x})$ at different layers of ResNet50.

| Source | Target | Cosine distance | | |
|---|---|---|---|---|
| | | I-FGSM | ILA | Aug-ILA |
| ResNet50 | VGG19 | 0.0110 | 0.0067 | 0.0089 |
| | Inception V3 | 0.0016 | 0.0010 | 0.0016 |
| Inception V3 | ResNet50 | 0.0024 | 0.0013 | 0.0017 |
| | VGG19 | 0.0042 | 0.0029 | 0.0026 |
| VGG19 | ResNet50 | 0.0049 | 0.0031 | 0.0035 |
| | Inception V3 | 0.0016 | 0.0013 | 0.0016 |

**Table 2:** Cosine distance between $\mathbf{x}'' - \mathbf{x}$ and $\nabla_{\mathbf{x}} J(\theta, \mathbf{x}, y)$. A larger cosine distance refers to a closer prediction of the gradient direction.

## 5    CONCLUSION

In this paper, we present Aug-ILA, which applies extra augmentation to the input references in the ILA framework. We evaluate different image transformations to be applied, and select random cropping to be our base augmentation. The transformation has to be applied consistently to all the input images in order to align the values of the intermediate feature maps. Moreover, we apply two more augmentations exploiting the adversarial perturbation, namely, reverse adversarial update and attack interpolation across iterations. By incorporating all the augmentations introduced, Aug-ILA not only outperforms ILA and its variants, but also the combination of state-of-the-art methods for transfer-based attacks. We have conducted extensive studies to study the effects of different hyper-parameters. Finally, we provide explanations on the effect of augmentation in terms of the weakening effect to the perturbation strength. As we only perform simple image transformation operations to augment the ILA references, one potential improvement is to consider more sophisticated augmentations such as automatic data augmentation, which will be marked for future studies.

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

# A EFFECT OF REVERSE ADVERSARIAL UPDATE ON THE MODEL PERFORMANCE

We would like to verify the intuition of using reverse adversarial update, specifically, whether in practice such operation can decrease the loss and increase the model confidence. Therefore, we randomly sample 5000 images, including those that are misclassified by the model. Then we apply I-FGSM10 in the form of reverse adversarial update to the images, and pass them back to the model for classification. We record the loss and apply softmax to the logits to obtain the confidence, and report the average result of all 5000 images. The result is summarized in Table 3.

Furthermore, we also inspect the class activation map (CAM) of the source model. Figure 4 visualizes some of the results. For some images (the first two), updating reversely helps the model focus on more features. For the remaining (the latter two), adversarial reverse update does not exhibit a significant change of intermediate layers' contribution to the ground-truth class.

| Model | Image | Loss | Confidence | Accuracy |
|-------|-------|------|------------|----------|
| ResNet50 | Clean | 0.9755 | 0.7913 | 75.08% |
| | Reversely updated | 0.3070 | 0.8847 | 92.20% |
| Inception V3 | Clean | 1.1165 | 0.7339 | 76.44% |
| | Reversely updated | 0.3525 | 0.8296 | 94.60% |
| VGG19 | Clean | 1.1431 | 0.7419 | 70.54% |
| | Reversely updated | 0.5756 | 0.8998 | 89.98% |

**Table 3:** Changes in loss and confidence after applying reverse adversarial update on the images for ResNet50.

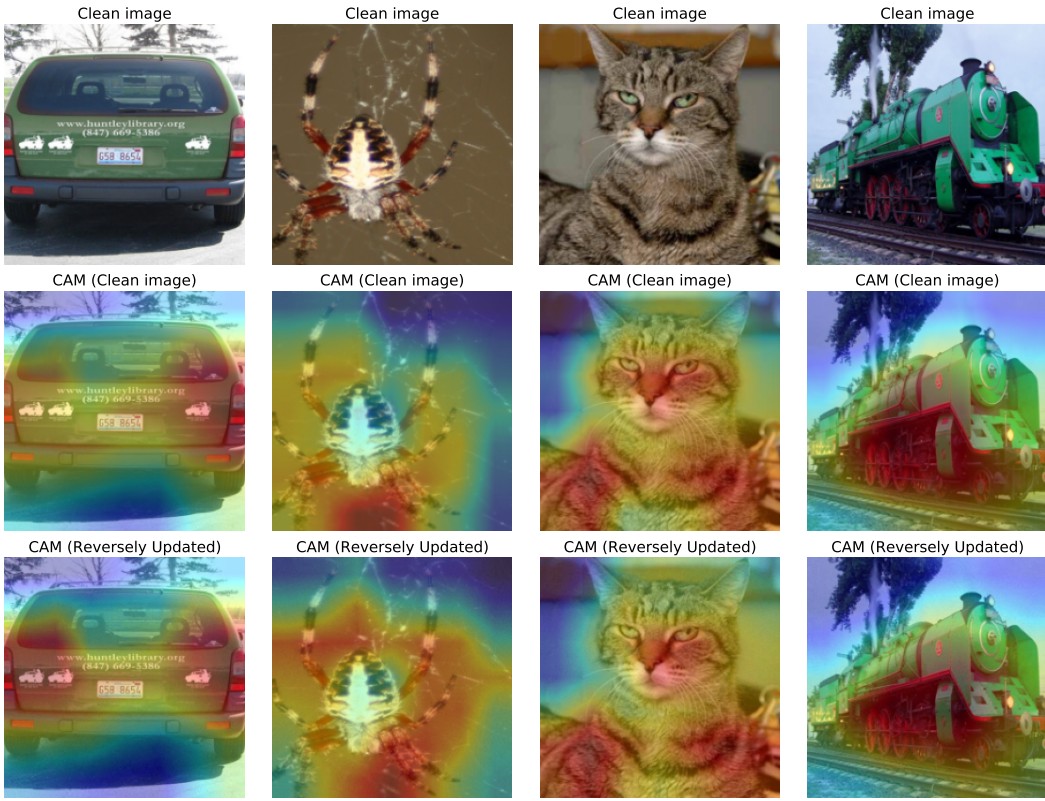

**Figure 4:** CAM visualization of the images after reverse adversarial update.

To test the effect of reversely updated examples on targeted models, we also feed them into the remaining undefended models and observe the change in accuracy. The result is reported in Table 4, which shows that reverse adversarial update not only benefits the source model, but also many of the other models with similar architecture. It shows that reverse adversarial update can be beneficial to many of the target models under proper choices of hyper-parameters such as the source model.

| Source model | Target model | | | | |
| | ResNet50 | Inception v3 | WRN | VGG19 | PNASNet |
|---|---|---|---|---|---|
| - | 75.08% | 76.44% | 77.58% | 70.54% | 71.34% |
| ResNet50 | 92.20% | 79.48% | 82.22% | 74.06% | 72.84% |
| Inception V3 | 78.58% | 94.60% | 80.70% | 73.62% | 72.88% |
| VGG19 | 78.46% | 79.56% | 80.06% | 89.98% | 71.76% |
| Source model | Target model | | | | |
| | DenseNet | ResNeXt | MobileNet | SENet | Average |
| - | 75.86% | 78.30% | 70.48% | 75.36% | 74.55% |
| ResNet50 | 80.70% | 82.70% | 73.70% | 80.70% | 79.84% |
| Inception V3 | 80.54% | 81.54% | 73.04% | 78.80% | 79.37% |
| VGG19 | 80.58% | 81.02% | 74.38% | 79.00% | 79.42% |

**Table 4:** Top-1 accuracy of the models under the example with reverse adversarial update from different source models. The source model "-" indicates clean example without any perturbation.

## B  EFFECT OF DIFFERENT IMAGE TRANSFORMATION OPERATIONS

For translation, we shift the image by a random factor of $[-0.1, 0.1]$ of the image size, both vertically and horizontally. Cropping operations consist of random cropping of $95\%$ of the size, followed by an upsampling to the original shape. The rotation degree ranges between $[-90°, 90°]$. Color jittering tweaks all the brightness, contrast, saturation by 0.2 and hue by 0.1. Random cropping is found to be the one resulting in the highest attack transferability, followed by translation and rotation. Pixel-wise transformations such as color jittering only exhibit limited improvement to transferability comparing to spatial transformations.

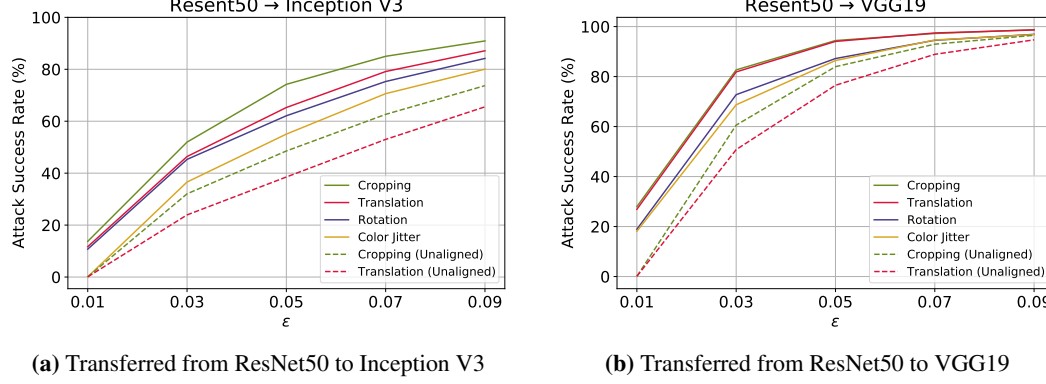

(a) Transferred from ResNet50 to Inception V3        (b) Transferred from ResNet50 to VGG19

**Figure 5:** Attack success rates of applying translation, cropping, rotation and color jittering as augmentation in ILA.

# C EFFECT OF REVERSE ADVERSARIAL UPDATE ON ATTACK TRANSFERABILITY

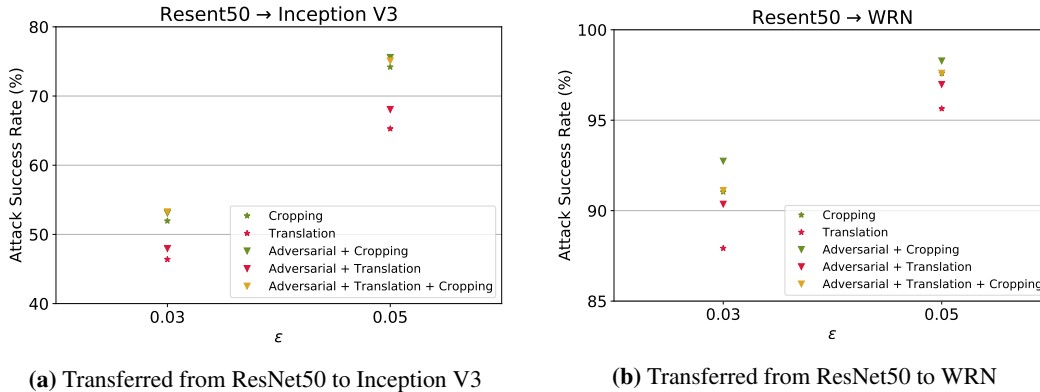

**(a)** Transferred from ResNet50 to Inception V3    **(b)** Transferred from ResNet50 to WRN

**Figure 6:** Attack success rates of different augmentation, in conjunction with adversarial reverse update (denoted as adversarial). The performance of simple augmentation without reverse adversarial update is also included as baseline.

# D HYPER-PARAMETERS USED IN THE BASELINES

We reproduce the algorithms of variance tuning, CTM (DIM, TIM, SIM), MI-FGSM and NI-FGSM according to the repository released by Wang & He (2021), and test the attacks under our experimental settings. The list of hyper-parameters used in the experiments is shown in Table 5, which are the default values in their corresponding papers. For the implementation of SGM and LinBP, we directly use the official implementation released by Guo et al. (2020).

| Method | Hyper-parameter | Value |
|---|---|---|
| MI-FGSM | $\alpha$ | 1.0 |
| NI-FGSM | $\alpha$ | 1.0 |
| DIM | Probability | 0.5 |
|  | Upscale ratio | 1.1 |
| TIM | Kernel size | $7 \times 7$ |
| SIM | Scale copies | $5\ (i = 0, 1, 2, 3, 4)$ |
| Variance Tuning | $N$ | 20 |
|  | $\beta$ | $1.5/255$ |
| SGM | $\lambda$ | 0.5 |
| LinBP | I-FGSM iteration | 300 |
|  | Layer | '3-1' for ResNet50 |

**Table 5:** Hyperparameters used in the baselines.

# E ATTACKS USING OTHER SOURCE MODELS

| $\epsilon$ | Method | ResNet50 | Inc. v3 | WRN | VGG19* | PNASNet |
|---|---|---|---|---|---|---|
| | I-FGSM + ILA | 77.32% | 45.42% | 74.78% | **99.40%** | 70.30% |
| 0.05 | I-FGSM + ILA++ | 79.56% | 47.08% | 76.04% | 99.30% | 70.76% |
| | I-FGSM + Aug-ILA (Ours) | **87.36%** | **66.40%** | **87.00%** | 99.02% | **82.34%** |
| | I-FGSM + ILA | 53.78% | 25.94% | 49.54% | **99.34%** | 46.30% |
| 0.03 | I-FGSM + ILA++ | 57.88% | 27.94% | 52.94% | 99.30% | 49.48% |
| | I-FGSM + Aug-ILA (Ours) | **66.22%** | **40.82%** | **64.90%** | 97.82% | **60.24%** |
| $\epsilon$ | Method | DenseNet | ResNeXt | MobileNet | SENet | Average |
| | I-FGSM + ILA | 67.38% | 61.68% | 85.34% | 72.90% | 72.72% |
| 0.05 | I-FGSM + ILA++ | 68.66% | 63.56% | 85.60% | 74.46% | 73.89% |
| | I-FGSM + Aug-ILA (Ours) | **81.72%** | **80.30%** | **93.06%** | **85.84%** | **84.78%** |
| | I-FGSM + ILA | 40.12% | 36.26% | 66.52% | 49.42% | 51.91% |
| 0.03 | I-FGSM + ILA++ | 44.88% | 39.70% | 68.70% | 53.62% | 54.94% |
| | I-FGSM + Aug-ILA (Ours) | **57.38%** | **54.02%** | **77.58%** | **63.32%** | **64.70%** |

* The source model used to generate the attack.

**Table 6:** Attack success rates of ImageNet adversarial examples on different models, generated from VGG19 in the untargeted setting.

| $\epsilon$ | Method | ResNet50 | Inc. v3* | WRN | VGG19 | PNASNet |
|---|---|---|---|---|---|---|
| | I-FGSM + ILA | 57.26% | 97.84% | 51.98% | 61.70% | 54.16% |
| 0.05 | I-FGSM + ILA++ | 57.80% | 97.60% | 51.72% | 60.54% | 52.86% |
| | I-FGSM + Aug-ILA (Ours) | **85.16%** | **98.14%** | **83.24%** | **89.16%** | **81.38%** |
| | I-FGSM + ILA | 34.18% | 96.98% | 29.62% | 37.56% | 34.02% |
| 0.03 | I-FGSM + ILA++ | 36.74% | **97.18%** | 31.90% | 38.90% | 35.14% |
| | I-FGSM + Aug-ILA (Ours) | **57.94%** | 91.18% | **53.54%** | **65.22%** | **56.38%** |
| $\epsilon$ | Method | DenseNet | ResNeXt | MobileNet | SENet | Average |
| | I-FGSM + ILA | 46.14% | 42.24% | 64.38% | 48.86% | 58.28% |
| 0.05 | I-FGSM + ILA++ | 47.42% | 44.08% | 62.62% | 49.72% | 58.26% |
| | I-FGSM + Aug-ILA (Ours) | **78.88%** | **74.54%** | **89.16%** | **80.42%** | **84.45%** |
| | I-FGSM + ILA | 27.00% | 23.72% | 43.36% | 29.64% | 39.56% |
| 0.03 | I-FGSM + ILA++ | 29.06% | 25.68% | 43.82% | 31.58% | 41.11% |
| | I-FGSM + Aug-ILA (Ours) | **47.80%** | **43.06%** | **65.96%** | **51.58%** | **59.18%** |

* The source model used to generate the attack.

**Table 7:** Attack success rates of ImageNet adversarial examples on different models, generated from Inception V3 in the untargeted setting.

## F    EXPERIMENTS WITH SINGLE-STEP ATTACK

| $\epsilon$ | Method | ResNet50[*] | Inc. v3 | WRN | VGG19 | PNASNet |
|---|---|---|---|---|---|---|
| 0.05 | FGSM | 80.86% | 28.78% | 37.08% | 46.24% | 36.04% |
| | FGSM + ILA | 93.36% | 28.56% | 51.94% | 58.22% | 38.48% |
| | FGSM + ILA++ | **97.88%** | 43.14% | 74.88% | 65.92% | 44.04% |
| | FGSM + Aug-ILA (Ours) | 96.98% | **57.48%** | **87.20%** | **91.76%** | **67.40%** |
| 0.03 | FGSM | 84.68% | 20.44% | 29.00% | 30.96% | 23.52% |
| | FGSM + ILA | 88.60% | 19.48% | 38.02% | 39.50% | 24.04% |
| | FGSM + ILA++ | **97.02%** | 29.82% | 60.02% | 50.64% | 30.60% |
| | FGSM + Aug-ILA (Ours) | 89.72% | **32.46%** | **65.44%** | **72.60%** | **40.08%** |

| $\epsilon$ | Method | DenseNet | ResNeXt | MobileNet | SENet | Average |
|---|---|---|---|---|---|---|
| 0.05 | FGSM | 32.48% | 30.72% | 45.32% | 41.42% | 42.10% |
| | FGSM + ILA | 39.10% | 40.12% | 52.42% | 52.58% | 50.53% |
| | FGSM + ILA++ | 65.80% | 65.48% | 66.12% | 74.12% | 66.38% |
| | FGSM + Aug-ILA (Ours) | **79.48%** | **74.20%** | **84.18%** | **83.70%** | **80.26%** |
| 0.03 | FGSM | 25.10% | 22.38% | 29.86% | 31.70% | 33.07% |
| | FGSM + ILA | 28.50% | 29.22% | 35.76% | 39.56% | 38.08% |
| | FGSM + ILA++ | 50.54% | **50.62%** | 52.66% | 61.36% | 53.70% |
| | FGSM + Aug-ILA (Ours) | **55.68%** | 48.22% | **58.74%** | **62.56%** | **58.39%** |

[*] The source model used to generate the attack.

**Table 8:** Attack success rates of ImageNet adversarial examples with single-step FGSM as the reference attack, generated from ResNet50 in the untargeted setting.

## G    EXPERIMENTS WITH MORE DATASETS

To show the generalization capability of Aug-ILA, we conduct experiments with two more datasets, namely CIFAR-10 and CIFAR-100 (Krizhevsky, 2012). Partly following Li et al. (2020b), we consider four models: VGG19 with batch normalization (Simonyan & Zisserman, 2015), Wide ResNet-28-10 (WRN) (Zagoruyko & Komodakis, 2016), ResNeXt-29 (ResNeXt) (Xie et al., 2017) and DenseNet-190 (DenseNet) (Huang et al., 2017). The model parameters are obtained from a public repository[3]. For both datasets, we randomly sample 3000 images from the test set that are classified correctly by all four models and we pick VGG19 to be the source model. The experimental results using CIFAR-10 and CIFAR-100 are shown in Table 9 and Table 10 respectively.

For both CIFAR-10 and CIFAR-100, the attack success rates of the baselines are much higher than that of ImageNet. Although Aug-ILA mostly attains the highest attack success rate among the baselines, we observe that the degree of improvement is not as high as that in Table 1. It may be because of the tiny size and low resolution ($32 \times 32$ for both CIFAR-10 and CIFAR-100) of the images, leading to a reduction in the effect of data augmentation such as random cropping.

| $\epsilon$ | Method | VGG19[*] | WRN | ResNeXt | DenseNet | Average |
|---|---|---|---|---|---|---|
| 0.05 | I-FGSM | 99.54% | 67.91% | 67.91% | 64.00% | 74.84% |
| | I-FGSM + ILA | 99.54% | 97.85% | 98.19% | 97.08% | 98.16% |
| | I-FGSM + ILA++ | **99.57%** | 98.00% | 98.17% | 97.40% | 98.29% |
| | I-FGSM + Aug-ILA | 98.77% | **98.22%** | **98.46%** | **98.06%** | **98.38%** |
| 0.03 | I-FGSM | 99.32% | 55.27% | 55.43% | 51.64% | 51.64% |
| | I-FGSM + ILA | **99.45%** | 88.26% | 89.61% | 87.15% | 91.12% |
| | I-FGSM + ILA++ | 99.13% | 89.67% | 90.13% | 87.40% | 91.58% |
| | I-FGSM + Aug-ILA | 98.31% | **93.05%** | **93.42%** | **91.45%** | **94.06%** |

[*] The source model used to generate the attack.

**Table 9:** Attack success rates of CIFAR-10 adversarial examples on different models, generated from VGG19 in the untargeted setting.

---

[3] https://github.com/bearpaw/pytorch-classification

| $\epsilon$ | Method | VGG19[*] | WRN | ResNeXt | DenseNet | Average |
|---|---|---|---|---|---|---|
| | I-FGSM | **99.45%** | 48.15% | 39.45% | 42.41% | 57.37% |
| 0.05 | I-FGSM + ILA | 99.03% | 92.73% | 88.97% | 88.63% | 92.34% |
| | I-FGSM + ILA++ | 99.10% | **93.13%** | 89.53% | 89.70% | **92.87%** |
| | I-FGSM + Aug-ILA | 96.08% | 92.65% | **90.56%** | **90.69%** | 92.50% |
| | I-FGSM | 98.84% | 38.91% | 31.94% | 33.77% | 50.87% |
| 0.03 | I-FGSM + ILA | **98.91%** | 80.77% | 72.71% | 73.93% | 81.58% |
| | I-FGSM + ILA++ | 98.53% | 81.23% | 74.77% | 74.60% | 82.28% |
| | I-FGSM + Aug-ILA | 95.22% | **84.37%** | **78.33%** | **78.56%** | **84.12%** |

[*] The source model used to generate the attack.

**Table 10:** Attack success rates of CIFAR-100 adversarial examples on different models, generated from VGG19 in the untargeted setting.

## H   EFFECT OF AUG-ILA ON DEFENDED MODELS

In this section, we evaluate the effect of Aug-ILA on defended models. We consider three robust models using ensemble adversarial training (Tramer et al., 2018), including an ensemble of 3 adversarially trained Inception V3 (Inc-v3$_{ens3}$), an ensemble of 4 adversarially trained Inception V3 (Inc-v3$_{ens4}$), and an ensemble of 3 adversarially trained Inception-ResNet-v2 (IncRes-v2$_{ens}$). On top of that, we also test the attacks against three defenses: HGD (Liao et al., 2018), R&P (Xie et al., 2018) and NIPS-r3[4], which are the top 3 defenses in the NIPS 2017 adversarial competition. For all three defenses, we adopt the default models and hyper-parameters used in their corresponding repositories.

We ported the implementation of Aug-ILA into the same experimental setup used in previous works (Dong et al., 2018; 2019; Wang & He, 2021), which includes 1000 sampled test images and the pretrained parameters of the models. Then we compare the baselines, including I-FGSM, MI-FGSM, MI-CT-FGSM, NI-CT-FGSM, and VNI-CT-FGSM, with ILA and Aug-ILA inserted. We pick Inception V3 as the source model, and all attacks are performed with perturbation size $\epsilon = 16/255 (\approx 0.0627)$. For ILA and Aug-ILA, we finetune the attack for 50 iterations by choosing the layer 6a in Inc-V3$_{adv}$, an adversarially trained Inception V3 model. The attack success rate against each of the defenses is reported in Table 11. From preliminary results, ILA cannot improve the attack success rate on the defended models, so it is removed in the experiments with stronger attacks. Nevertheless, Aug-ILA is able to improve the attack success rate of most of the attacks, except VNI-CT-FGSM.

| Attack | Inc-v3$_{ens3}$ | Inc-v3$_{ens4}$ | IncRes-v2$_{ens}$ | HGD | R&P | NIPS-r3 |
|---|---|---|---|---|---|---|
| I-FGSM | 12.1% | 10.9% | 5.80% | 2.70% | 4.00% | 8.30% |
| I-FGSM + ILA | 10.1% | 10.6% | 5.00% | 0.10% | 2.50% | 7.90% |
| I-FGSM + Aug-ILA | 43.0% | 33.4% | 24.0% | 35.1% | 23.6% | 14.3% |
| MI-FGSM | 14.1% | 13.0% | 6.60% | 4.60% | 5.00% | 8.30% |
| MI-FGSM + ILA | 14.1% | 12.5% | 6.40% | 0.70% | 3.60% | 5.40% |
| MI-FGSM + Aug-ILA | 44.2% | 36.3% | 24.6% | 33.6% | 25.3% | 28.5% |
| MI-CT-FGSM | 65.5% | 62.1% | 45.5% | 56.6% | 44.5% | 52.5% |
| MI-CT-FGSM + Aug-ILA | 69.0% | 65.8% | 49.8% | 62.1% | 51.6% | 65.0% |
| NI-CT-FGSM | 58.8% | 54.4% | 40.0% | 49.2% | 38.0% | 46.1% |
| NI-CT-FGSM + Aug-ILA | 66.8% | 62.4% | 44.5% | 55.8% | 46.5% | 52.8% |
| VNI-CT-FGSM | 79.1% | 77.4% | 65.3% | 72.7% | 63.5% | 70.8% |
| VNI-CT-FGSM + Aug-ILA | 75.6% | 73.2% | 58.3% | 68.4% | 81.6% | 64.0% |

**Table 11:** Attack success rates of ImageNet adversarial examples, generated from Inception V3 in the untargeted setting.

---

[4] https://github.com/anlthms/nips-2017/tree/master/mmd

# I  HYPER-PARAMETERS FOR IMAGE TRANSFORMATION OPERATIONS

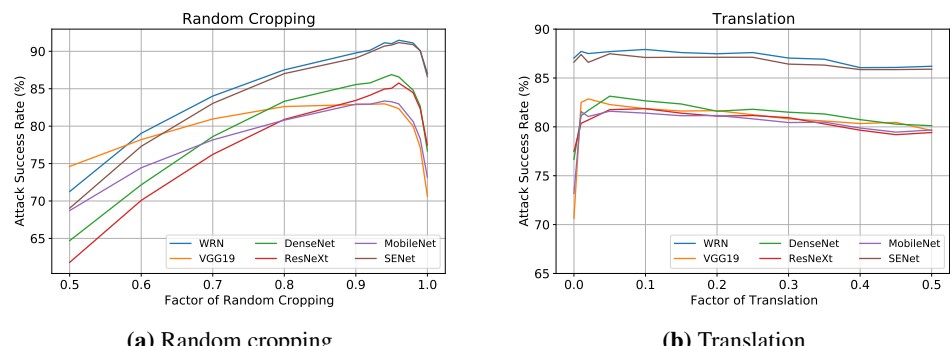

**(a)** Random cropping

**(b)** Translation

**Figure 7:** Attack success rates of different hyper-parameters for random cropping and translation, using ResNet50 as the source model with $\epsilon = 0.03 (\approx 8/255)$.

# J  EFFECT OF $\alpha$ ON ATTACK INTERPOLATION

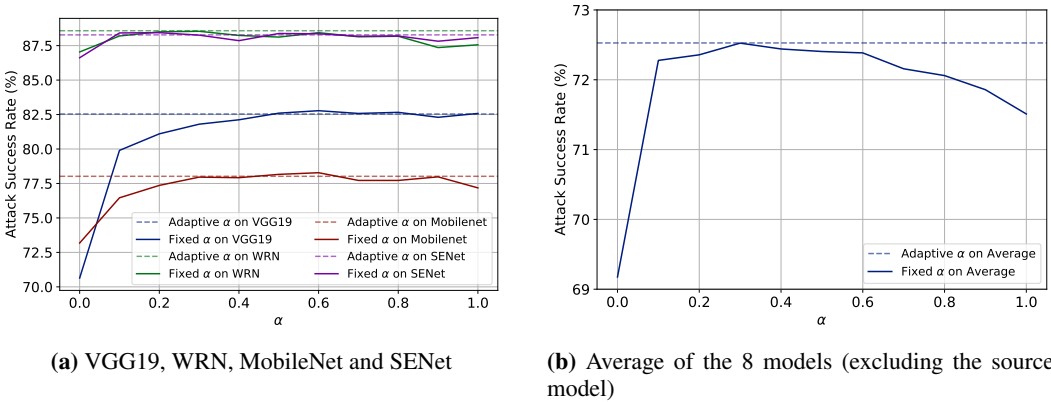

**(a)** VGG19, WRN, MobileNet and SENet

**(b)** Average of the 8 models (excluding the source model)

**Figure 8:** Attack success rates over different values of $\alpha$, using ResNet50 as the source model with $\epsilon = 0.03$.

# K  INTERMEDIATE FEATURE DISCREPANCIES ON OTHER MODELS

We extend the experiments examining the L2 norm of $F_l(\mathbf{x}'') - F_l(\mathbf{x})$ in Figure 3. Using the attacks generated by the same source model (ResNet50), we also test the magnitude of the intermediate feature discrepancies for VGG19 and Inception V3, as shown in Figure 9. The magnitude of the norm depends on the size of the feature maps, so it cannot be compared directly across layers but relatively among different attacks. Agreeing with our explanation in Section 4.4, the L2 norm of Aug-ILA starts with comparable strength to that of ILA in shallow layers, but outpaces ILA when the layers get deeper.

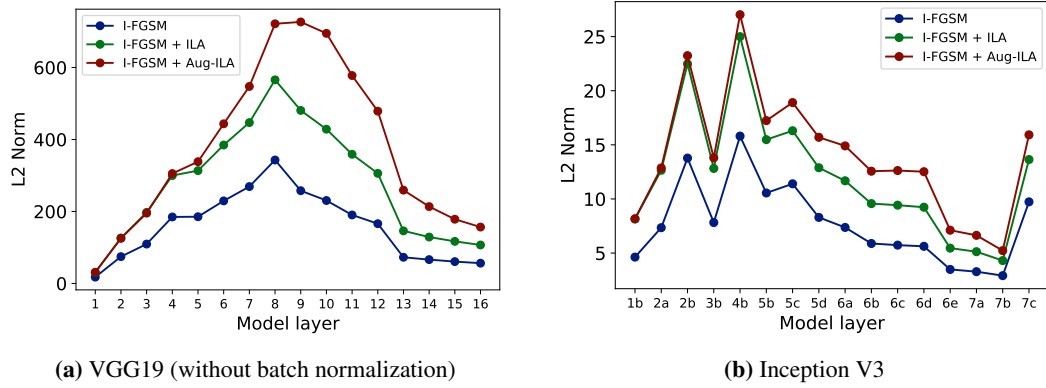

**(a)** VGG19 (without batch normalization)    **(b)** Inception V3

**Figure 9:** L2 norm of $F_l(\mathbf{x}'') - F_l(\mathbf{x})$ at the intermediate layers of VGG19 and Inception V3.

## L   ABLATION STUDY ON THE THREE AUGMENTATIONS IN AUG-ILA

Aug-ILA can be regarded as a generalization of the original ILA, with the extension of three major parameters. With the cropping size set to 1.0, $\alpha = 0$, and reverse adversarial update disabled, Aug-ILA degenerates to ILA. We conduct ablation study on the effect of the three augmentations by removing each of them from Aug-ILA and reporting the performance of the reduced methods in Table 12. Alternatively, we also observe the change in attack success rates with each of the three augmentation components added into the original ILA, and report the result in Table 13.

It is found that cropping contributes the most to attack transferability, followed by the remaining two methods. However, from Table 13, we can see that attack interpolation alone does not contribute much unless it is applied together with other augmentations.

| $\epsilon$ | Method | ResNet50* | Inc. v3 | WRN | VGG19 | PNASNet |
|---|---|---|---|---|---|---|
| | I-FGSM + Aug-ILA | 99.78% | **90.02%** | **99.04%** | 98.76% | **93.18%** |
| 0.05 | w/o Cropping | 99.78% | 66.78% | 96.92% | 95.12% | 75.84% |
| | w/o Attack interpolation | **99.96%** | 77.10% | 98.00% | 94.60% | 82.22% |
| | w/o Reverse adversarial update | 99.74% | 89.20% | 98.54% | **98.80%** | 91.40% |
| | I-FGSM + Aug-ILA | 99.42% | **66.02%** | **93.92%** | **92.90%** | **75.76%** |
| 0.03 | w/o Cropping | 99.72% | 39.74% | 86.64% | 79.80% | 49.76% |
| | w/o Attack interpolation | **99.96%** | 54.26% | 91.82% | 83.46% | 62.22% |
| | w/o Reverse adversarial update | 98.76% | 62.56% | 91.46% | 92.28% | 71.38% |
| $\epsilon$ | Method | DenseNet | ResNeXt | MobileNet | SENet | Average |
| | I-FGSM + Aug-ILA | **98.40%** | **96.90%** | **98.12%** | **98.72%** | **96.99%** |
| 0.05 | w/o Cropping | 93.96% | 92.68% | 93.26% | 96.98% | 90.15% |
| | w/o Attack interpolation | 96.52% | 95.74% | 94.92% | 97.68% | 92.97% |
| | w/o Reverse adversarial update | 98.22% | 96.80% | 97.80% | 98.44% | 96.55% |
| | I-FGSM + Aug-ILA | **91.04%** | **87.76%** | **90.38%** | **93.40%** | **87.84%** |
| 0.03 | w/o Cropping | 77.06% | 76.08% | 75.66% | 85.96% | 74.49% |
| | w/o Attack interpolation | 87.26% | 85.92% | 83.52% | 92.06% | 82.28% |
| | w/o Reverse adversarial update | 90.18% | 86.32% | 88.86% | 90.94% | 85.86% |

* The source model used to generate the attack.

**Table 12:** Comparison of the attack success rates of ImageNet adversarial examples on different models, with each augmentation removed from Aug-ILA.

| $\epsilon$ | Method | ResNet50* | Inc. v3 | WRN | VGG19 | PNASNet |
|---|---|---|---|---|---|---|
| | I-FGSM + ILA | 100.0% | 51.50% | 93.20% | 86.60% | 65.20% |
| | w/ Attack interpolation | 99.84% | 61.88% | 96.82% | 95.42% | 72.12% |
| 0.05 | w/ Reverse adversarial update | 99.94% | 61.36% | 96.72% | 89.38% | 65.94% |
| | w/ Cropping | 99.96% | 74.30% | 97.42% | 94.62% | 78.78% |
| | w/ All (Aug-ILA) | 99.78% | 90.02% | 99.04% | 98.76% | 93.18% |
| | I-FGSM + ILA | 99.98% | 34.56% | 79.88% | 66.72% | 43.32% |
| | w/ Attack interpolation | 99.60% | 34.22% | 85.06% | 78.04% | 43.14% |
| 0.03 | w/ Reverse adversarial update | 99.98% | 37.86% | 87.46% | 71.76% | 42.72% |
| | w/ Cropping | 99.96% | 50.80% | 90.14% | 82.26% | 56.30% |
| | w/ All (Aug-ILA) | 99.42% | 66.02% | 93.92% | 92.90% | 75.76% |
| $\epsilon$ | Method | DenseNet | ResNeXt | MobileNet | SENet | Average |
| | I-FGSM + ILA | 85.50% | 91.00% | 84.00% | 91.90% | 83.21% |
| | w/ Attack interpolation | 73.12% | 73.14% | 74.04% | 83.28% | 88.96% |
| 0.05 | w/ Reverse adversarial update | 93.36% | 92.28% | 92.68% | 96.22% | 87.08% |
| | w/ Cropping | 92.68% | 92.20% | 89.26% | 96.20% | 91.68% |
| | w/ All (Aug-ILA) | 98.40% | 96.90% | 98.12% | 98.72% | 96.99% |
| | I-FGSM + ILA | 69.16% | 78.48% | 67.60% | 80.16% | 68.87% |
| | w/ Attack interpolation | 73.12% | 73.14% | 74.04% | 83.28% | 71.52% |
| 0.03 | w/ Reverse adversarial update | 78.08% | 77.82% | 74.36% | 87.52% | 73.06% |
| | w/ Cropping | 83.74% | 82.34% | 82.04% | 89.18% | 79.64% |
| | w/ All (Aug-ILA) | 91.04% | 87.76% | 90.38% | 93.40% | 87.84% |

* The source model used to generate the attack.

**Table 13:** Comparison of the attack success rates of ImageNet adversarial examples on different models, with each augmentation added to Aug-ILA.

## M    STUDY ON THE RUNNING TIME OF THE ALGORITHMS

In this section, we compare the running time between ILA and Aug-ILA. We report the running time required to run attack on the 5000 sampled images from the ILSVRC2012 validation set. The setup is the same as the previous experiments, which we run I-FGSM for 10 iterations followed by ILA for 50 iterations. The batch size is set to 100. All the experiments are performed on a machine with an Intel Xeon Silver 4214R CPU and Nvidia GeForce RTX3090 GPU. The result is summarized in Table 14.

| Method | Running Time (s) | |
|---|---|---|
| | Total | Average (per batch) |
| I-FGSM | 336 | 6.72 |
| I-FGSM + ILA | 947 | 18.94 |
| I-FGSM + Aug-ILA | 1083 | 21.66 |

**Table 14:** Running time comparison between ILA and Aug-ILA.

Comparing with ILA, Aug-ILA has three more augmentations applied. Firstly, random cropping is performed once for every ILA iteration. Secondly, reverse adversarial update requires basic arithmetic on the images. The computation time of these two processes is insignificant and hence negligible. However, attack interpolation requires the computation of $\alpha$ with the norms. Since we need to obtain the intermediate feature output of the new $\mathbf{x}''$, an extra forward pass to the model is required per iteration. Such extra pass is the major overhead introduced by Aug-ILA. From Table 14, the 50 extra forward passes increases the running time by around $14\%$, which is still reasonable in terms of generation of adversarial attack.

# N    MORE VISUALIZATIONS ON THE GENERATED EXAMPLES

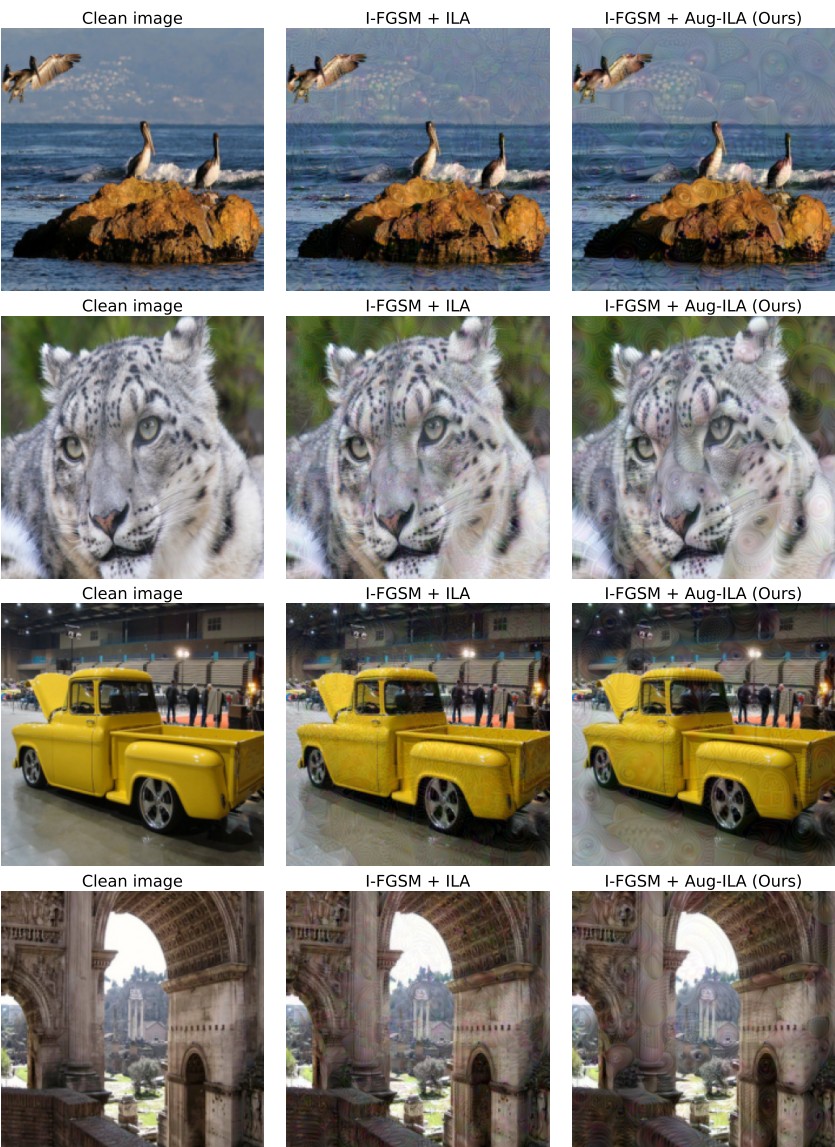

**Figure 10:** Visualization of the generated images among: clean image, ILA, and Aug-ILA (Ours), with the perturbation budget $\epsilon = 0.03 (\approx 8/255)$.

