# OpenReview forum: "Aug-ILA: More Transferable Intermediate Level Attacks with Augmented References"
_ICLR.cc/2022/Conference — ICLR 2022 Submitted_

### Official Review · Reviewer_TcRw · 2021-10-30

**Correctness:** 2
**Technical Novelty And Significance:** 3
**Empirical Novelty And Significance:** 2
**Recommendation:** 3
**Confidence:** 4

**Details Of Ethics Concerns:**

N.A.

**Main Review:**

Strengths
1.	The paper is well written, and the content is clear, except some minor problems.
2.	The three techniques effectively enhance ILA.
3.	I appreciate the use of reverse adversarial update on the clean example and the automatic parameter selection scheme.


Weaknesses (major)
1.	Recently, most of transferability methods published in top conferences are evaluated on both undefended and defended models. The authors only evaluated their method on undefended models. If we consider that the evaluation of defended models is a must, the paper is not ready for publish.
2.	Some strong transferability methods, such as LinBP, have evaluated on target attacks. However, no such experiment was provided.
3.	The method is very sensitive to the selection of layers in the source model. For example, the Inception V3 in Fig. 2(c) shows that the difference of the attack success rates between the 6a and 6e layer is more than 20% and the VGG19 in Fig. 2(b) shows that the difference of the attack success rates between 9 and 12 layer is around 20%. Once non-optimal layers are selected, the attack performance would drop very significantly. Does the proposed method still outperform other state-of-the-art methods in that situation? In the real black attacks, attackers cannot select the layers, as the authors did, because they don’t have attack accuracy of target models.
4.	The authors mentioned “VMI-SI-TI-DI-FDSM does not work well in terms of black-box attack, despite its remarkable performance in attacking defended models”. The statement has two problems. 1) VMI-SI-TI-DI-FDSM was examined on a black-box setting when their method was evaluated on defended model. 2) Why don’t compare the proposed method with it on the defended models?


Weaknesses (others)
1.	The authors method that “The combination of LinBP and SGM also incurs overhead as it requires 300 iterations of I-FGSM on ImageNet, which is 30 times more than the reference attack in Aug-ILA. Thus, statement is inaccurate. In setup, the authors mention that they use 10 iterations for I-FGSM and 50 iterations for Aug -ILA. In the 50 iterations, gradients are needed. Thus, this statement is inaccurate, although speed is not very important for adversarial attack.
2.	In Eq. 1, better to write clearly that dot represents dot product and the outputs of Eq. 2 and 3 are vectors.
3.	Appendix F, in the training, do the victim models use copping with 0.9-0.97 as their augmentation? Thus, the curves have similar behaviours.
4.	No experiment shows that the proposed method can work with other methods to enhance overall transferability.



**Summary Of The Paper:**

The authors proposed / used three techniques to enhance the transferability of ILA on undefended models. The techniques include image augmentation, reverse adversarial update on the clean example and reference attack update through interpolation with an automatic parameter selection scheme. The authors evaluated their method on nine undefended models and showed the proposed method outperforms ILA and ILA++ and other state-of-the-art methods

**Summary Of The Review:**

The paper is well-written and the proposed method does improve ILA. However, due to the insufficient experiments, especially evaluation on the defended models and target attacks and the sensitivity of the selected layers, this paper is not ready for publish.

---

> ### Author Response · Authors · 2021-11-22
> **The Response to Reviewer TcRw (Part 1 / 2)**
>
> We would like to thank the reviewer for the feedback. Below is the response to the comments.
>
> **Weaknesses (major)**
>
> > 1. The authors only evaluated their method on undefended models. If we consider that the evaluation of defended models is a must, the paper is not ready for publish.
>
> We respectfully disagree with the reviewer’s comment that considering “evaluation of defended models is a must”. Transferability between models and overcoming defense are two separate tasks that are not necessarily tied together. Studying attack transferability suggests the common decision boundary shared by different models. By crafting a highly transferable attack, the attack “generalizes” to different models. However, breaking defense focuses more on the study of denoising/anti-denoising. The main goal of Aug-ILA (and other ILA-based frameworks) is to transfer between architectures, which belongs to the former study. The attack generated is expected to contain the common (mis)interpretation of the classes. Whether such interpretation is able to survive through the denoisers is considered to be another work.
>
> However, we agree that evaluation of defended models can be very beneficial to understand the characteristics of the generated attacks. Therefore, we conduct experiments on defended models and report the result in Appendix H. A quick summary is that Aug-ILA exhibits apparent improvement to the attack success rate on defended models with most attacks, in contrast to the performance decay of ILA. For some particularly strong attacks such as VNI-CT-FGSM, adding Aug-ILA slightly worsens the performance on defended models, but still achieves the highest success rate among undefended models.
>
> Again, we would like to reiterate that breaking defense is not the major focus of our work, and should not be treated as importantly as that in transferring between model architectures.
>
>
> > 2. Some strong transferability methods, such as LinBP, have evaluated on target attacks. However, no such experiment was provided.
>
> As argued by [1], transfer-based attacks work very poorly in the targeted setting, especially when the number of classes is large (such as ImageNet). From the result in LinBP [2], the average attack success rate of targeted LinBP + PGD (the strongest method in Table 12) is 6.74%, 0.98%, 0.06% under perturbation size of 16/255, 8/255 and 4/255, respectively. In addition, none of the targeted attacks generated by Inception V3 gets larger than 1% (Table 13 in [2]). The general performance is too poor and we can hardly draw meaningful conclusions from the result. That is why we do not conduct experiments with targeted attacks.
>
> For smaller datasets such as CIFAR-10 and MNIST, transfer-based attacks may work better in the targeted setting. However, the generation of targeted transfer-based attacks is still discouraged generally.
>
> > 3. The method is very sensitive to the selection of layers in the source model. In the real black attacks, attackers cannot select the layers, as the authors did, because they don’t have attack accuracy of target models.
>
> From the result, the choice of the layer is consistent with ILA & ILA++. Note that the result shown in Figure 2 is the average attack success rate over 8 models. Grounded by the target models, these selections should be reasonable enough as an initial attempt in real black-box settings. From all the source models tested, selecting layers close to the middle always gives optimal or suboptimal attack success rate. Therefore we generally encourage choosing the layers in the middle, specifically 3-1 for ResNet50, layer 9 for VGG, layer 6a for Inception V3, etc. as the default choice.
>
> > 4. The authors mentioned “VMI-SI-TI-DI-FDSM does not work well in terms of black-box attack, despite its remarkable performance in attacking defended models”. The statement has two problems. 1) VMI-SI-TI-DI-FDSM was examined on a black-box setting when their method was evaluated on defended model. 2) Why don’t compare the proposed method with it on the defended models?
>
> The statement means VMI-SI-TI-DI-FDSM is less remarkable in transferring “between our selected architectures'. The focus is not on the word “black-box setting” but “under our selected architecture”. We have fixed the wording to make it clearer in the updated version.

---

> ### Author Response · Authors · 2021-11-22
> **The Response to Reviewer TcRw (Part 2 / 2)**
>
> **Weaknesses (others)**
>
> > 1. The authors method that “The combination of LinBP and SGM also incurs overhead as it requires 300 iterations of I-FGSM on ImageNet, which is 30 times more than the reference attack in Aug-ILA. Thus, statement is inaccurate. In setup, the authors mention that they use 10 iterations for I-FGSM and 50 iterations for Aug -ILA. In the 50 iterations, gradients are needed. Thus, this statement is inaccurate, although speed is not very important for adversarial attack.
>
> In this context, reference attack means I-FGSM (without ILA). This is mentioned to highlight that Aug-ILA does not require a large number of I-FGSM iterations, as the attack will be finetuned later. We agree that it causes confusion with “Aug-ILA being 30 times faster”, and we have replaced this sentence with a more detailed discussion on the running time of Aug-ILA in Appendix M of the revised version.
>
> >2. In Eq. 1, better to write clearly that dot represents dot product and the outputs of Eq. 2 and 3 are vectors.
>
> Thank you for the suggestion. We have adjusted the description of equations 1-3 in Section 2.2.
>
> > 3. Appendix F, in the training, do the victim models use copping with 0.9-0.97 as their augmentation? Thus, the curves have similar behaviours.
>
> Most of the models are pre-trained by PyTorch (torchvision), except SENet and PNASNet which are obtained from other repositories. The PyTorch ones use RandomResizedCrop to transform into 224x224 during training.
> In our experiments, the short dimension of the images is scaled to 256, followed by a 224x224 center cropping. Then the 224x224 image is further randomly cropped into 0.9-0.97 in the original Appendix F. Since multiple cropping/resize are involved, the proportion of the images cropped is unlikely to align with square cropping as we performed in the experiments.
>
> > 4. No experiment shows that the proposed method can work with other methods to enhance overall transferability.
>
> Aug-ILA is formulated as a fine-tuning method for a reference attack. By nature, it must be combined with some attacks to function. In our revised version, we also added single-step FGSM as the reference attack (in Appendix F). In the experiments with defended models (in Appendix H), we used different SOTA as the reference attack to show that Aug-ILA can enhance the overall transferability for most of the attacks.
>
>
> ## References
> [1] Yanpei Liu, Xinyun Chen, Chang Liu and Dawn Song. Delving into Transferable Adversarial Examples and Black-box Attacks. In ICLR, 2017
>
> [2] Yiwen Guo, Qizhang Li, and Hao Chen. Backpropagating linearly improves transferability of adversarial examples. In NeurIPS, 2020

---

> ### Comment · Reviewer_TcRw · 2021-12-07
> **After rebuttal**
>
> I have read the responses from the authors and the changes in the revised paper. The followings are my comments.
>
>
> Weakness (Main)
>
> When all the recent transferable attack methods published in top conferences are examined on defended models, generally, the authors are expected to do the same. Thus, I don’t agree the authors’ responses.
> The results in Table 11 (Appendix H) show that the proposed method generally improves other methods on defended models. However, it degrades the results from the best method, VNI-CT-FGSM. It means that the best result is achieved without the proposed method.
> For the target attack question, I don’t have comment on authors’ response although I still believe that it is good to test on the target setting.
> The authors only address the issues about selection of layers partially. For any new source models, someone needs to rerun the experiments to select the best layers. The comparison is slightly unfair. The proposed method and other ILA methods need more than one model to select the layer parameter. However, some other methods really can perform attack based on single source model. In response, the authors recommend choosing the layers in the middle. However, according to VGG result in Fig. 2, the performance fluctuation is still very large.
>
> Weakness (others)
>
> I accept all the responses to the authors.
>
> I keep my original score mainly due to the experiments on defended models and selection of layer.

---

### Official Review · Reviewer_uNtx · 2021-11-02

**Correctness:** 3
**Technical Novelty And Significance:** 3
**Empirical Novelty And Significance:** 3
**Recommendation:** 6
**Confidence:** 3

**Main Review:**

- The overall structure of this paper is clear, and  extensive experiment results are provided to support their ideas.
- Based on the observation from the  previous work, diversity transformation and perturbation interpolation are used to improve the transferability. The experimental results show that the augementation is simple and effective. However, I still have some doubts: how does the adversarial perturbation generated against the transformed examples correspond to the original ones? Especially random cropping with large size causes great damage to the image, making the generated perturbation is extremely incosistent with the original image in pixel. But the adversarial example shown in Figure 1 is natural.
- The author expect the reverse adversarial update input can highly activate the intermediate layers and provide a better attack direction guidence. However, the reduction of classification loss in the Appendix A seems obvious.
Is it possible to show the change of feature attention map after the first update?
- Although the three basic methods proposed are effective and intuitive, there is a problem to be solved: how to choose the index of the attack layer, which seems to have no prior guidance other than referring to existing papers.
- Beyond experiment results, the author provide explanations on the effect of augmentation in terms of the weakening effect to the perturbation strength.

**Summary Of The Paper:**

This paper presents a transfer-based attack method based on Intermediate Level Attack(ILA). Image augementation and reverse adversarial updateare applied to ILA input, to make diverse adversarial references. Moreover, the interpolation of cumlative attacks can maintain a better transfer direction.The paper is clear and well-written and experimental results and analysis support the claims.

**Summary Of The Review:**

In summary, I find the experiments satisfying, although the additional experiment show can make the claims more convincing.

---

> ### Author Response · Authors · 2021-11-22
> **The Response to Reviewer uNtx**
>
> We would like to thank the reviewer for the feedback. Below is the response to the comments.
>
> > How does the adversarial perturbation generated against the transformed examples correspond to the original ones?
>
> For a single Aug-ILA iteration, due to cropping, only a portion of the pixels is fed into the model and updated. Since Aug-ILA is performed for 50 iterations, the images are randomly cropped 50 times, so different parts of the images can be covered and updated. In the original Appendix F (Appendix I of the revised version), random cropping works the best with the size at around 0.95 (i.e., 5% of the image is cropped), showing that cropping the image too much can also harm transferability.
>
>
> > Rev update: Obvious reduction in loss. Is it possible to show the change of feature attention map after the first update?
>
> Thank you for the suggestion. We have added CAM visualization in Appendix A, demonstrating the change of feature attention when a reversely updated image is fed into the model.
>
> > How to choose the index of the attack layer, which seems to have no prior guidance other than referring to existing papers.
>
> We start the experiments with insight from ILA++ [1] (in which the layers were chosen empirically), which is 3-1 for ResNet, 9 for VGG, etc. In Figure 2, we reported the average attack success rates on different target models over the choice of layer. The result exhibits a similar shape compared to that of existing paper, which suggests that the peak performance of Aug-ILA is very similar to ILA and ILA++. Therefore, the same guidance in previous works also applies to ours. In general, for all the models we have tested, selecting a layer close to the middle always gives performance close to optimal. Hence we suggest doing so for unknown target models.
>
>
> ## Reference
> [1] Qizhang Li, Yiwen Guo, and Hao Chen. Yet another intermediate-level attack. In ECCV, 2020

---

### Official Review · Reviewer_2dLg · 2021-11-02

**Correctness:** 3
**Technical Novelty And Significance:** 2
**Empirical Novelty And Significance:** 3
**Recommendation:** 5
**Confidence:** 3

**Main Review:**

strengths:
===========
- Using more number of reference attacks seems to be a valid solution (as also done in ILA++), and using data augmentation techniques to achieve this goal is logical, providing a simple solution with significant improvements.

- The empirical results consistently show significant improvements to other approaches, specially to closely related methods such as ILA and ILA++.

- The study of hyperparameters supports the choices made

- Ablation result demonstrates the effectiveness of each data augmentation

weaknesses:
===========

## lack of comparison with published results:

- The configurations used (e.g, perturbation size, dataset combinations, architecture type) in the reported experiments are different from those used in the published results, and it appears all results are reported from experiments carried out by the authors. This raises some concerns on whether the baselines were used justly, with best hyperparameters etc. For example ILA++ [Li et at., 2020] reports results on CIFAR100, ILA [Huang et al,. 2019] on CIFAR10 and ImageNet (but different models than used in this work). Why isn't the models/dataset combinations used in related work (e.g, in [Wang et al., 2021], [Wu et al., 2021]) aren't used? Also it is unclear which methods from [Wang et al., 2021.] are dubbed as VMI-SI-DI-TI-FGSM, and what are the related details regarding how they relate to VMI-SI-DI-TI-FGSM ? Also it seems that in  [Wang et al., 2021.] VNI-FGSM performs better than VNI-FGSM. What's the reason for choosing VMI-FGSM over VNI-FGSM? As for I-FGSM + LinBP + SGM + ILA, and LinBP from [Wu et al., 2020], it is unclear what method from [Wu et al., 2020] is I-FGSM + LinBP + SGM + ILA referring to.



- The proposed method's empirical performance is only tested on one dataset, and its generalization capabilities onto other datasets with the tuned hyperparameters is not demosntrated. Providing results on other datasets (e.g, CIFAR100) would demonstrate generalisation on their approach. Especially, because the best parameters chosen, are found via the results conducted on the validation set of Imagenet, the same set as model is evaluated, and there os no held-out set to account for overfitting.

- Only  $L_{inf}$  I-FGSM attacks are used, and for example, the effectiveness by using l2, single-step FGSM, was not demonstrated (as done in [Wu et al., 2020]. Also the choices of perturbation sizes are not justified.

- All results are only reported on undefended models, and it is unclear how these attacks would transfer if an adversarially-robust model was used as target (as done in [Wu et al., 2021.].

- results reported only on a single run, and there is no performance report on the effect of random seed. As in [Wu et al., 2020], it is recommended to report mean and std over several runs of the algorithm to demonstrate stability of the attack's convergence.


## limitation of ablation studies:

- In ablations reported in Appx. I, results when ignoring a single factor removed; and the two other augmentations are in place, are reported. Based on these, it appears that removing cropping has the largest effect. I would like to know what is the performance of ILA only with each of the augmentations (cropping, reverse update, interpolation). This way we can pinpoint the contribution of each individual part. At the moment, we can only observe a result of combined steps.


## computational cost:

Please discuss the computational complexity of the proposed method in comparison with SOTA and baselines used. E.g, how much additional computation will these 3 data augmentations require compared to ILA?


additional comments:
====================

- in  $L_{inf}$  attacks, it is common to state perturbation size by $\epsilon= x/255$. I recommend providing the perturbation size in this format, or at least clarify this point.

- on the reverse adversarial update, it seems that this step will move the sample further away from the decision boundary.

- connection to related work: it would be insightful to discuss connections to related work (ILA, ILA++). Can Aug-ILA be considered a generalisation of any of the previous methods? or a special case? or there is no major connection?

- some theoretical analysis on the convergence/generalisation on Aug-ILA, could shed light on the functionality of such approaches, and is encouraged if it could provide insights. For example, ILA, ILA++, and Aug-ILA could be studied under a simplified setting.


refs:
=====

[Wu et al., 2021]: Improving the Transferability of Adversarial Samples with Adversarial Transformations

[Wang et al., 2021]: Enhancing the Transferability of Adversarial Attacks through Variance Tuning

[Wu et al., 2020]: Skip Connections Matter: On the Transferability of Adversarial Examples Generated with ResNets





**Summary Of The Paper:**

This paper introduces improvements over Transferable Intermediate-level Attacks (ILA), by incorporating data augmentation into the attack tuning stage. The data augmentations introduced include 3 kinds: simple transformations (cropping), reverse-adversarial update, and attack interpolation. An empirical study has been conducted on Imagenet, and 3 well-known architectures have been used as source models, and the attack transferability was evaluated onto 9 models using I-FGSM with $L_{inf}$ distance and 2 different perturbation sizes.
Ablation studies have been conducted to demonstrate the effectiveness of hyperparameter choices, and different kinds of data augmentation used, demonstrating the effectiveness of the proposed approach compared to other ILA-based blackbox attacks.



**Summary Of The Review:**

The proposed method is simple, and provides significant improvements. However, I have the following concerns:

- More clarity on some of the baselines and evaluation used is required (see above).

- There is possibility of overfitting on hyperparameters due to using a single dataset for evaluations. Providing additional results will help resolve this issue.

- The generality of the proposed method is not sufficiently demonstrated (only one attack, only one dataset).

Please see full comments above, and address if possible.


After rebuttal:
=====

After the rebuttal, results on 2 new datasets, as well as results on defended models, and single-step attacks are provided, comparing AugILA with other ILA variants, which partially addresses my concerns regarding these points. Additionally, some concerns regarding the use of baselines have been resolved by providing responses to my comments.

The additional results shows that the proposed method consistently and in the majority of the cases, improves over ILA and ILA++, and I am convinced that the proposed approach improves over ILA/ILA++.
However, unfortunately no comparison is provided with other SOTA methods beyond ILA-based approaches, which limits the paper's contributions.
More specifically, my comments regarding evaluation on new datasets, single-step attacks, and defended models are not fully addressed due to the fact that the provided results are only comparisons to ILA-based approaches (ILA, ILA++), and baselines such as VMI-CT-FGSM and I-FGSM + LinBP + SGM + ILA, which were among the baseline presented in Table 1 have been missed.
Especially, because as reported in Table 1, VMI-CT-FGSM and I-FGSM + LinBP + SGM + ILA have better success rates than of the ILA baseline.
This leaves uncertainty on how the proposed method would perform on other datasets, single-step attacks, and defended models, compared to non-ILA approaches. In my opinion, this is the main drawback of the new results.

In summary, although I believe the paper has been improved after the rebuttal, due to the limitations mentioned above, I keep my score.
Nevertheless, I believe by strengthening the baseline comparisons, this work has the potential to become a valuable contribution for the community.

---

> ### Author Response · Authors · 2021-11-22
> **The Response to Reviewer 2dLg (Part 1 / 2)**
>
> We would like to thank the reviewer for the detailed feedback. Below is the response to the comments.
>
> ### lack of comparison with published results
>
> >This raises some concerns on whether the baselines were used justly, with best hyperparameters etc.
>
> For the reproduction of the baselines, we reproduce the algorithm based on their repositories.The hyper-parameters used are reported in Appendix D, which are the default values mentioned in the corresponding papers. We have also added more details in Appendix D to clarify the configuration used.
>
> > Why isn't the models/dataset combinations used in related work (e.g, in [Wang et al., 2021], [Wu et al., 2021]) aren't used?
>
> Since our work is based on ILA, ILA [4] and ILA++ [1] should be followed with top priority. Specifically, ILA uses ResNet18, SENet18, DenseNet121 and GoogLeNet (Inception V1) as the models, which seems to be a bit outdated and some of the models may not be deep enough. ILA++ uses **ResNet50**, **VGG-19**, ResNet-152, **Inception V3**, **DenseNet**, **MobileNet v2**, **SENet**, **ResNeXt**, **WRN**, **PNASNet** and MNASNet on ImageNet. Among the 11 models used in ILA++, we followed 9 models (as **highlighted**).
>
> For the combination used in Wang et al. [5] and Wu et al. [6], they used a slightly different setup, in which most of the model parameters and test datasets are originated from the NIPS 2017 adversarial competition. Within the two (ILA++ and NIP2017), we follow ILA++. For the choice of dataset, we choose ImageNet because it is commonly used in all the previous works.
>
> In the updated draft, we also adopt the setup of NIPS2017 in Appendix H on the evaluation of Aug-ILA against defended models.
>
> > Also it is unclear which methods from [Wang et al., 2021.] are dubbed as VMI-SI-DI-TI-FGSM, and what are the related details regarding how they relate to VMI-SI-DI-TI-FGSM ?
>
> VMI-SI-DI-TI-FGSM refers to the combination of Composite Transformation Method (CTM) and variance tuning, also with momentum iterative FGSM. In [Wang et al., 2021.], it is known as VMI-CT-FGSM. We expand CT (CTM) into the actual methods SIM + DIM + TIM in order to provide a clearer indication on the involvement of each method. We have edited the text to use the original name VMI-CT-FGSM instead.
>
> > Also it seems that in [Wang et al., 2021.] VNI-FGSM performs better than VMI-FGSM. What's the reason for choosing VMI-FGSM over VNI-FGSM?
>
> In our experiments reproducing the works, VMI-SI-DI-TI-FGSM (VMI-CT-FGSM) performs slightly better than VNI-SI-DI-TI-FGSM (VNI-CT-FGSM). Specifically, for $\epsilon = 0.03~(8/255)$, the average attack success rate of VNI-SI-DI-TI-FGSM is 62.78%, compared to 64.15% for VMI-SI-DI-TI-FGSM (VMI-CT-FGSM). We report the best one in Table 1 and thus VMI-CT-FGSM is used.
>
> > As for I-FGSM + LinBP + SGM + ILA, and LinBP from [Wu et al., 2020], it is unclear what method from [Wu et al., 2020] is I-FGSM + LinBP + SGM + ILA referring to.
>
> The name I-FGSM + LinBP + SGM + ILA comes from LinBP [2] but not Wu et al. [6], referring to running I-FGSM with LinBP and SGM enabled, followed by ILA for 100 iterations. In [2], it was called LinBP+I-FGSM+ILA+SGM. The order is slightly different, but the method is strictly the same.
>
> > Providing results on other datasets (e.g, CIFAR100) would demonstrate generalisation on their approach
>
> Thank you for the suggestion. The main reason for sticking to ImageNet is because it is the most common dataset used in the previous work. We have also added experimental results on CIFAR10 and CIFAR100 in Appendix G.
>
> > Only L-inf I-FGSM attacks are used, but not L-2 or single-step FGSM, and for example, the effectiveness by using l2, single-step FGSM, was not demonstrated
>
> Thank you for the comment. It is not common to use L-2 constraint in the study for I-FGSM (otherwise it will become FGM) and its extended attack. It is more well-known to be used in CW and probably other black-box attacks. On the other hand, single-step attacks are not strong enough and less effective to be applied together with the ILA framework. However, we agree that including more reference attacks shows the generalization of Aug-ILA. Therefore, we have added the result on single-step FGSM in Appendix F.
>
>
> > Also the choices of perturbation sizes are not justified.
>
> The choice of perturbation size is based on previous works. Since the perturbation size used in ILA is not consistent, we follow ILA++ [1] and LinBP [2], where 0.03 (8/255) and 0.05 (13/255) are used. In [2], 0.1 (25.5/255) is also used. However, we believe setting 0.1 as the perturbation size is too large for an adversarial attack since it is easily noticeable by humans, and therefore 0.1 is excluded from our experiments.

---

> ### Author Response · Authors · 2021-11-22
> **The Response to Reviewer 2dLg (Part 2 / 2)**
>
> > All results are only reported on undefended models, and it is unclear how these attacks would transfer if an adversarially-robust model was used as target
>
> Thank you for the suggestion. We have added experiments on defended models in Appendix H. However, we would like to highlight that Aug-ILA focuses mainly on transferring attacks between model architectures, rather than mitigating defense. In our formulation, the target of Aug-ILA is never intended to bypass defense, thus the result on defended models should not be treated as important as the result on different architectures.
>
> > Results reported only on a single run, and there is no performance report on the effect of random seed.
>
> While we believe reporting the effect of random seed is a plus, we do not prioritize it compared to other modifications. Wu et al. [3] changed the random seed in order to resample the 5000 images, while their proposed algorithm does not incur much randomness. Although we have a higher degree of randomness due to random cropping, repeating it 50 times (with 50 ILA iterations) can result in a much stabler condition. Moreover, the major focus of a transfer-based attack should not be the precision of the success rate, but the relative comparison between attacks. We find such steps are much less significant than the others, and thus do not report it in the paper.
>
> ---
>
> ### limitation of ablation studies
>
> > I would like to know what is the performance of ILA only with each of the augmentations (cropping, reverse update, interpolation).
>
> We can still figure out the influence of each modification by considering the difference between the full version (Aug-ILA) and the single wo/ version, but we are happy to provide more studies on each of the components. We have added one more table in Appendix L (originally Appendix I), showing the changes with the increment of each component.
>
> ---
>
> ### computation cost
>
> > How much additional computation will these 3 data augmentations require compared to ILA?
>
> We have added the evaluation on the computation time in Appendix M. To quickly mention here, Aug-ILA is approximately 14% slower than ILA, due to the extra forward pass required in the computation of $\alpha$ in attack interpolation. The remaining processes (random cropping, basic arithmetic on images) are insignificant and hence negligible.
>
> ---
>
> ### additional comments:
>
> > I recommend providing the perturbation size in this format, or at least clarify this point.
>
> Thank you for the suggestion. We have also added the format of $x/255$ in the text where the perturbation size is mentioned. For the tables, we keep the original format in order to align better with previous works.
>
> > on the reverse adversarial update, it seems that this step will move the sample further away from the decision boundary.
>
> We have conducted more study on the effect of reverse adversarial update. The results are included in Appendix A. Hopefully it can address some of the concerns.
>
> > Can Aug-ILA be considered a generalisation of any of the previous methods? or a special case? or there is no major connection?
>
> Aug-ILA can be regarded as a generalization of the original ILA, with the extension of three major
> parameters. With the cropping size set to 1.0, $\alpha = 0$, and reverse adversarial update disabled,
> Aug-ILA degenerates to ILA. We have also added it in Appendix L (Ablation Study) of the revised version.
>
> > Some theoretical analysis on the convergence/generalisation on Aug-ILA, could shed light on the functionality of such approaches, and is encouraged if it could provide insights
>
> Due to the difficulty in representing the effect of various augmentations to the model features, a theoretical analysis can be a major undertaking. We will pursue the analysis of ILA/Aug-ILA as a separate study in the future.
>
> ---
>
> ### References
> [1] Qizhang Li, Yiwen Guo, and Hao Chen. Yet another intermediate-level attack. In ECCV, 2020
>
> [2] Yiwen Guo, Qizhang Li, and Hao Chen. Backpropagating linearly improves transferability of adversarial examples. In NeurIPS, 2020
>
> [3] Dongxian Wu, Yisen Wang, Shu-Tao Xia, James Bailey, and Xingjun Ma. Skip connections matter: On the transferability of adversarial examples generated with resnets. In ICLR, 2020
>
> [4] Qian Huang, Isay Katsman, Horace He, Zeqi Gu, Serge Belongie, and Ser-Nam Lim. Enhancing
> adversarial example transferability with an intermediate level attack. In ICCV, 2019
>
> [5] Xiaosen Wang and Kun He. Enhancing the transferability of adversarial attacks through variance
> tuning. In CVPR, 2021
>
> [6] Weibin Wu, Yuxin Su, Michael R. Lyu, and Irwin King. Improving the transferability of adversarial
> samples with adversarial transformations. In CVPR, 2021

---

> ### Comment · Reviewer_2dLg · 2021-11-29
> **After rebuttal**
>
> After the rebuttal, results on 2 new datasets, as well as results on defended models, and single-step attacks are provided, comparing AugILA with other ILA variants, which partially addresses my concerns regarding these points. Additionally, some concerns regarding the use of baselines have been resolved by providing responses to my comments.
>
> The additional results shows that the proposed method consistently and in the majority of the cases, improves over ILA and ILA++, and I am convinced that the proposed approach improves over ILA/ILA++.
> However, unfortunately no comparison is provided with other SOTA methods beyond ILA-based approaches, which limits the paper's contributions.
> More specifically, my comments regarding evaluation on new datasets, single-step attacks, and defended models are not fully addressed due to the fact that the provided results are only comparisons to ILA-based approaches (ILA, ILA++), and baselines such as VMI-CT-FGSM and I-FGSM + LinBP + SGM + ILA, which were among the baseline presented in Table 1 have been missed.
> Especially, because as reported in Table 1, VMI-CT-FGSM and I-FGSM + LinBP + SGM + ILA have better success rates than of the ILA baseline.
> This leaves uncertainty on how the proposed method would perform on other datasets, single-step attacks, and defended models, compared to non-ILA approaches. In my opinion, this is the main drawback of the new results.
>
> In summary, although I believe the paper has been improved after the rebuttal, due to the limitations mentioned above, I keep my score.
> Nevertheless, I believe by strengthening the baseline comparisons, this work has the potential to become a valuable contribution for the community.

---

### Official Review · Reviewer_JiJd · 2021-11-03

**Correctness:** 3
**Technical Novelty And Significance:** 2
**Empirical Novelty And Significance:** 2
**Recommendation:** 6
**Confidence:** 4

**Main Review:**

**Strengths**

1. This paper is well organized.
2. The proposed AUG-ILA algorithm is simple, practical, and effective. Experiments on various datasets show that it outperforms both the original ILA algorithms and other state-of-the-art methods with a large margin.

**Weakness**
1. Lack of novelty. Although the experiment results are impressive, the proposed AIG-ILA algorithm is a very practical extension. No new theories or methods about ILA are proposed.
2. In Section 3.1, this paper claims that the reverse adversarial update is used to boost the confidence of clean image $x$. The update formulation is $x − (x^\prime− x) = x − \epsilon$ where $\epsilon$ is the adversarial
perturbation. Although $x − \epsilon$ will decrease the loss for the original model, it doesn’t always decrease the loss of the target model. So the reverse adversarial update operation that will boost confidence might not be well supported unless the transferability of adversarial examples is proved.
3. This paper argues that reverse update operation will help to get more useful attack information from discrepancy feature maps. It is better to show some visualization results to support this view.

**Summary Of The Paper:**

The paper proposed the augmented Intermediate Level Attack (ILA) algorithm to strengthen the transferability of adversarial examples. Also, it claimed that increasing the diversity of input references could improve the generalization of adversarial examples when attacking different models. Specifically, this paper performs augmentation operations, including common image data augmentations and transformations exploiting adversarial perturbation (e.g., reverse adversarial updates and attack interpolation on the reference attack), before
maximizing the projection of intermediate feature map discrepancies.

**Summary Of The Review:**

Generally, this paper proposes a simple and practical extension of the ILA algorithm. The experiments on various benchmarks are strong. However, this paper lacks the support of theory and some views might not be well supported.

---

> ### Author Response · Authors · 2021-11-22
> **The Response to Reviewer JiJd**
>
> We would like to thank the reviewer for the feedback. Below is the response to the comments.
>
>
> > 1. Lack of novelty. No new theories or methods about ILA are proposed.
>
> Although Aug-ILA can be viewed as a combination of multiple existing methods (except reverse update), the greatest contribution is the discovery that image augmentation works particularly well with the ILA framework. The effectiveness of the method surpasses other works with similar studies on transformation on the images/image gradient such as DIM [1], TIM [2], NI/SIM [3], etc. Besides, with the augmentation in Aug-ILA, visualization of the generated output is also interesting (Figure 1 and Appendix N in the updated version), which seems to induce an obvious confusion to human eyes on its fine texture rather than uninterpretable noise.
>
>
> > 2. Rev adversarial update doesn’t always decrease the loss of the target model
>
> We agree that there is no proof/guarantee on decreasing the loss of the target model. In fact, typical transfer-based adversarial attacks are also not proven to be effective on arbitrary target models. The effect of decreasing loss and increasing confidence on the source model acts as an insight to apply reverse adversarial update, as well as using transformations involving the adversarial perturbation. From the ablation study (in Appendix L of the revised version), we can see that applying reverse adversarial update effectively increases the attack success rate, verifying its usefulness empirically.
>
> To provide more study on the effect of reverse adversarial update, we conducted two new experiments.
> 1. We added visualization of class activation map (CAM) on the normal and reversely updated examples.
> 2. We added a report on the change in accuracy by inputting a reversely updated example to the target models.
>
> Both experiments are reported in Appendix A. We hope the additional experiments can address your concern regarding the reverse adversarial update.
>
> > 3. Show visualization on the effect of reverse update
>
> Thank you for the suggestion. We have included the CAM visualization of the intermediate layer in Appendix A (Figure 4), as mentioned in (2).
>
>
> ## References
> [1] Cihang Xie, Zhishuai Zhang, Yuyin Zhou, Song Bai, Jianyu Wang, Zhou Ren, and Alan L Yuille. Improving transferability of adversarial examples with input diversity. In CVPR, 2019
>
> [2] Yinpeng Dong, Tianyu Pang, Hang Su, and Jun Zhu. Evading defenses to transferable adversarial examples by translation-invariant attacks. In CVPR, 2019
>
> [3] Jiadong Lin, Chuanbiao Song, Kun He, Liwei Wang, and John E Hopcroft. Nesterov accelerated gradient and scale invariance for adversarial attacks. In ICLR, 2020

---

> > ### Comment · Reviewer_JiJd · 2021-11-30
> > **After rebuttal**
> >
> > Thanks for the authors' response. The extra experiments (Tabel 4) proved that the inputs being updated reversely in the source model could increase the confidence in the target model. The authors only provided the CAM of feeding inputs into the source models. It would be better for me to see similar CAMs of target models.
> >
> > Generally, the proposed Aug-ILA is practical, effective, and intuitive but lacks theoretical contribution. Most of my concerns have been well addressed. I think the merit and flaw are both clear in the current version, I will change my score to weak accept accordingly.

---

### Author Response · Authors · 2021-11-22
**Updates and General Response**

We would like to thank all the reviewers for their valuable comments and suggestions. We have incorporated changes according to the reviewers’ comments, especially regarding the diversity of the experiments (more attacks, more datasets, evaluation on defended models, etc.).

Here is a quick summary of the major updates (all changes are highlighted in ${\color{blue}{\text{blue}}}$ in the revised version):
- Section 2.2: We adjusted the description of the operation and variable type in equations 1-3.
- Section 4.2: We adjusted the wordings and added references to the new experiments in the appendices.
- Appendix A: We modified Table 3 with the new setup, added CAM visualization on the images after reverse adversarial update and the accuracy comparison of the target models with reversely updated input.
- Appendix D: We added the description of reproducing the baselines.
- [NEW] Appendix F: We added experiments with single-step attack (FGSM).
- [NEW] Appendix G: We added experiments with CIFAR-10 and CIFAR-100.
- [NEW] Appendix H: We added experiments with defended models.
- Appendix L: In the ablation study, we added the incremental comparison (w/).
- [NEW] Appendix M: We added a study on the running time.
- Overall: We rearranged the order of appendices to make it more comprehensible. Also, we clarified the perturbation size in the format of $x/255$.

We hope the revised version could address most of the concerns raised by the reviewers. We appreciate any further inquiry and feedback. Hereby we would like to thank all the reviewers and the ACs again for reviewing the papers.

---

### Decision · Program_Chairs · 2022-01-20

**Decision:**

Reject

**Comment:**

This paper develops a new method, named Augmented Intermediate Level Attack (Aug-ILA), to improve the transferability of black-box attacks. Specifically, the proposed Aug-ILA contains three key modules: image transformations, reverse-adversarial update, and attack interpolation.

Overall, the reviewers think it is an interesting paper, but are concerned that the original ablations are not enough to support the effectiveness of the proposed method, including missing strong baseline attacks and defense methods, and only one dataset is considered. During the discussion period, the authors actively provide new results. However, the Reviewer TcRw and the Reviewer 2dLg are not fully convinced by the rebuttal, especially regarding 1) in these additional experiments, no comparison is provided with other SOTA attacks beyond ILA-based approaches; 2) Table 11 shows the proposed method even degrades (rather than improves) the performance of VNI-CT-FGSM on defense models;  3) the attack rate of the proposed method is sensitive to the selection of layers, therefore, need to be carefully tuned in experiments (which could lead to unfair comparisons to other attacks). These concerns are indeed legitimate, and should be addressed carefully before publication.

I encourage the authors to incorporate all the reviewers' comments and make a stronger submission next time.